# Automatic data-driven design and 3D printing of custom ocular prostheses

Johann Reinhard [1,2] ✉, Philipp Urban[1,3], Stephen Bell[4,5], David Carpenter [6] & Mandeep S. Sagoo [5,7,8]

Millions of people require custom ocular prostheses due to eye loss or congenital defects. The current fully manual manufacturing processes used by highly skilled ocularists are time-consuming with varying quality. Additive manufacturing technology has the potential to simplify the manufacture of ocular prosthetics, but existing approaches just replace to various degrees craftsmanship by manual digital design and still require substantial expertise and time. Here we present an automatic digital end-to-end process for producing custom ocular prostheses that uses image data from an anterior segment optical coherence tomography device and considers both shape and appearance. Our approach uses a statistical shape model to predict, based on incomplete surface information of the eye socket, a best fitting prosthesis shape. We use a colour characterized image of the healthy fellow eye to determine and procedurally generate the prosthesis's appearance that matches the fellow eye. The prosthesis is manufactured using a multi-material full-colour 3D printer and postprocessed to satisfy regulatory compliance. We demonstrate the effectiveness of our approach by presenting results for 10 clinic patients who received a 3D printed prosthesis. Compared to a current manual process, our approach requires five times less labour of the ocularist and produces reproducible output.

Loss of the eye causes vision deficit and a visible difference in appearance[1]. Reasons for loss of the eye include trauma, painful blind eye and eye tumours that cannot be treated conservatively[2]. After evisceration or enucleation of the eye, the orbital volume is made up from an orbital implant[3], placed under the conjunctiva. In enucleation the muscles can be attached to the implant or the conjunctival fornices, whereas evisceration in non-tumour cases retains the normal sclera of the eye, with the muscles retained in their normal position. A prosthetic eye is worn over the conjunctiva, held by the eyelids, and the movement from the implant is transmitted to the prosthesis due to action of the extraocular muscles (Fig. 1). Pre-manufactured stock eyes are selected to best match the patient, while custom ocular prostheses are manufactured specifically to match the patient's eye socket shape and companion eye's appearance.

Approximately 0.1% of the world's population wears a prosthetic eye[4]. Rehabilitation of patients after eye loss or potentially disfiguring surgery is important for cosmesis and psychological acceptance. It needs to fulfil a number of requirements; replacement of orbital volume, timely prosthetic manufacture once the socket heals, acceptable comfort, cosmesis and movements are all components that need to be addressed.

[1]Fraunhofer Institute for Computer Graphics Research IGD, Darmstadt, Germany. [2]Department of Computer Science, Technical University Darmstadt, Darmstadt, Germany. [3]Department of Computer Science, Norwegian University of Science and Technology, Gjøvik, Norway. [4]Ocupeye Ltd., Kenilworth, UK. [5]NIHR Biomedical Research Centre for Ophthalmology at Moorfields Eye Hospital and UCL Institute of Ophthalmology, London, UK. [6]Ocular Prosthetics Department, Moorfields Eye Hospital NHS Foundation Trust, London, UK. [7]Ocular Oncology Service, Moorfields Eye Hospital NHS Foundation Trust, London, UK. [8]Retinoblastoma Service, Royal London Hospital, Barts Health NHS Trust, London, UK. ✉e-mail: johann.reinhard@igd.fraunhofer.de

We developed an automatic, digital end-to-end process for the manufacture of ocular prosthetics. It uses minimally invasive optical coherence tomography (OCT) with a conformer to capture the anatomical topography of the anophthalmic socket, as well as the anatomy and coloration of the fellow normal eye. A data-driven design software automatically computes a digital 3D model of the ocular

prosthetic which is produced with a multi-material 3D printer. Our contribution is an automatic end-to-end approach to design and manufacture custom aesthetic prosthesis considering both shape fitting and appearance matching.

The current manufacture of custom ocular prostheses is a bespoke artisan and highly skilled manual process[5–7] requiring years of training, each prosthesis is hand crafted for an individual patient. In ocular prosthetic clinics, such as the one at Moorfields Eye Hospital NHS Foundation Trust (MEH) in London, they are made from polymethyl methacrylate (PMMA). The ocularist makes an alginate impression of the patient's eye socket (Fig. 2a) to cast a wax shape that he fits into the socket (Fig. 2b) and adjusts by removing or adding wax. The iris is painted onto a flat disc that is embedded, together with a clear PMMA cornea unit, into the wax shape to line up with the patient's fellow eye. This takes about 2 h with the patient in the clinic. White PMMA is packed with the painted iris into a plaster mould cast from the wax model and cured under pressure in a 6-h heat cycle. A thin layer is ground away from the front surface, staining and veining is added to the white scleral areas with threads of red yarn and a brush (Fig. 2c). The ground off layer is replaced with clear PMMA and after curing the prosthesis is polished, some adjustments of the shape are necessary (Fig. 2d) during supply to the patient (Fig. 2e). It requires approximately 8 h of manual labour to make one prosthesis[8,9]. As this is a hand-made artisan process many variables may be incorporated, and one ocularist treating the same patient twice may end up with slightly different shapes and appearances.

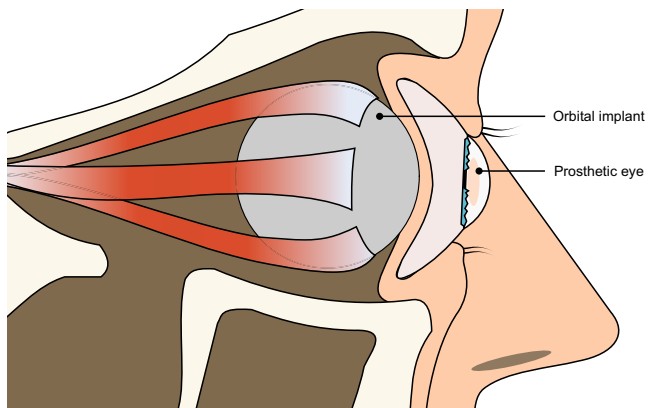

**Fig. 1 | Simplified schematic of an ocular prosthetic in the patient.** An orbital implant with the muscles attached is usually implanted in the eye socket after enucleation or evisceration and is covered by the conjunctiva. The prosthetic eye sits between the eye lids and the implant; it can be moved due to friction with the orbital implant.

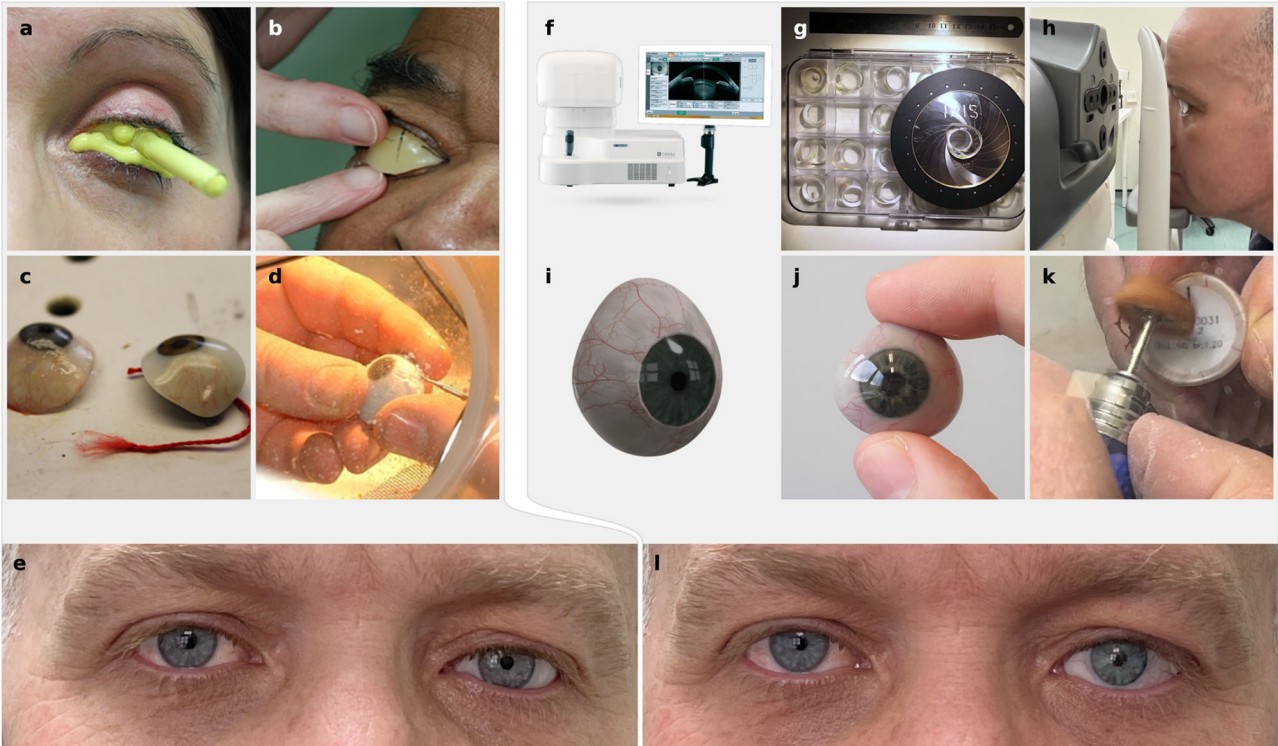

**Fig. 2 | Traditional manufacture of ocular prosthesis using PMMA and the digital end-to-end process. a** Traditionally the ocularist makes an alginate impression of the patient's eye socket, which is used to create a wax model. **b** The wax model is inserted into the patient's eye socket and adjusted to give the shape of the prosthesis. **c** After manufacture using PMMA the prosthesis is painted by hand using paintbrushes and coloured yarn. **d** The prosthesis shape is adjusted to be fitted to the patient during the supply visit. **e** Patient wearing the traditionally manufactured hand-made prosthesis (left eye, on the right in the image). **f** In our presented method an AS-OCT device, which is modified with a colour camera and

LED lights, is used to capture the patient data. For the scan of the eye socket the ocularist selects a conformer from a set (**g**) which is inserted into the eye socket during the scan (**h**). **i** Digital model of the prosthesis, the shape and appearance are automatically created using the OCT and colour image data. **j** Multi-material 3D printed prosthesis after post-processing. **k** The shape of the prosthesis is adjusted by the ocularist during the supply where necessary. **l** Digitally designed and produced prosthesis in Patient 4, who has lost the left eye (right in the image); it provides a better or at least similar colour and appearance match for the iris and sclera compared to the manually made prosthesis (**e**).

Another traditional method is glassblowing of white cryolite glass and use of coloured glasses to create iris and veins. The prosthesis is shaped by manipulating the heated glass; fit and appearance is assessed in the patient's socket after cooling. Compared to PMMA it can look more lifelike and is supplied in one appointment. However, it cannot be adjusted to the patient's socket surface meso-topography, is more delicate and can break, and must be replaced more frequently due to its surface becoming etched and rough.

In recent years additive manufacturing has gained significant attention in ophthalmology[10–13], two areas of interest are the manufacture of orbital and ocular prostheses[14]. Aside from shape and material properties, which are crucial elements for orbital prostheses[15–19], the appearance plays a key role in the acceptance of ocular prostheses by patients. Various approaches have been proposed to manufacture prosthetic eyes using 3D printing[9,20–28], but none describe a digital, automated, and data-driven design process that considers both shape and appearance. Instead of an automatic shape prediction based on OCT images, the shape is manually designed in CAD or 3D modelling software based on a CT scan of a wax model[20], a 3D-scan of a prosthesis[9,27], 3D-scans of impression moulds[22,23] and a manually set parametric model[25], an average prosthesis shape[21], or manually segmented cone beam computed tomography data[24,26]; while an automated shape design process using volume difference reconstruction[28] uses computed tomography that have an elevated cancer risk in patients with a cancer predisposition. None describes an automatic appearance recreation, either appearance is not considered[24–26], hand painted[20,28], or based on manually processed digital images[21–23] with colour calibration[9] or manual addition of veining[27]. All have a small samples size and were tested on none[9,21,27], one[26,28], two[20], or three patients[22] or three cadavers[24].

## Results

### Digital end-to-end process

In our digital end-to-end process, as shown in Fig. 2f–k, we scan the patient's fellow eye and eye socket with a modified standalone prototype Casia 2 (Tomey Corporation, Japan) anterior segment optical coherence tomography[29] (AS-OCT) device (Fig. 2f) that includes a colour camera. These devices are commonly used in clinical practice for imaging the anterior of the eye and have been proven to be useful to capture the socket surface[30]. It uses a low power optical laser with a wavelength of 1310 nm for imaging instead of x-rays, avoiding ionizing radiation and hence the risk of cancer.

The ocularist selects from a set of 12 conformers (Fig. 2g) the one that best fits in the patient's eye socket and inserts it (Fig. 3a). The conformer keeps the eyelids open, provides a shape bias for the determination of the prosthesis shape, and its window serves as a reference point for the extraction and alignment of the socket surface. The ocularist must ensure that it matches the socket surface and volume well when selecting the conformer, otherwise deformation of the socket tissue can provide erroneous socket surface information. The patient's head is fixated by a chin and head rest, the OCT head is positioned using a joystick to scan the eye socket (Fig. 2h). The scan itself takes about 2.4 s. Under controlled, dark lighting conditions the fellow eye of the patient is scanned using the same device, while an integrated colour camera captures a colour image of the eye. The ocularist may hold the eyelids apart to ensure that the whole iris is visible.

The captured data, consisting of the volumetric image stack of the eye socket (Fig. 3d), a mesh of the iris surface (Fig. 3i) and the raw colour image (Fig. 3m), is exported with the CASIA 2 software of the OCT device. The ocularist selects two veining parameters, thickness *th* and branching ratio *br*, by comparing the patient's eye with a set of examples veining patterns (Supplementary Fig. 5p). The data acquisition takes less than 30 min per patient.

From this data our data-driven design software automatically determines a fitting prosthesis shape and uses the colour image to generate a textured 3D model (Fig. 2i) which is then produced with a multi-material 3D printer in full colour (Fig. 2j, k). The ocularist may have to adjust the shape before supplying it to the patient (Fig. 2l).

### Design of the prosthesis shape

To determine a prosthesis shape we use the OCT scan data of the patient's anophthalmic socket. Since the surface of the socket is partly obscured by the eyelids that block the optical laser of the OCT only the central socket region is visible. To predict a shape that best fits the visible socket surface and fills the unseen volume we use a statistical shape model (SSM). These have been used successfully in a wide range of medical applications[31,32]. The dataset to build our model consists of a selection of 173 manually made ocular prostheses sourced from the MEH.

The prostheses are marked with a pattern to indicate orientation and position of the iris, scanned with a MEDIT T500 (Medit Corp., South Korea) dental 3D scanner, and aligned such that the limbi lie in the x-y-plane, are centred in the origin, and have the same orientation (Supplementary Fig. 1a–d). For each aligned scan mesh we compute a set of 838 corresponding points in the 3-dimensional space by performing a ray-marching procedure on the depth maps computed from its orthographic z-projection (Supplementary Fig. 1e–l). A common set of faces creates a closed surface representation.

Using singular value decomposition[33] we compute a principal component analysis[34] based statistical shape model that maps a shape parameter $\mathbf{x}$ to a mesh $S(\mathbf{x})$. We use the first 17 modes, so that the model explains 98% of the shape variance. The model replicates the data with a mean error of 0.27 (±0.06) mm, further statistical analysis can be found in the methods and Supplementary Fig. 2a–d.

The conformers are derived from the SSM based on 12 prostheses representing the most common shapes, each prosthesis is scanned and aligned, the corresponding surface points are projected in the shape space for the shape parameters $\mathbf{x}_c \in \mathbb{R}^{17}, c \in \{c_1, \ldots, c_{12}\}$. The conformer's base shape $S(\mathbf{x}_c)$ is modified by cutting off the front and creating a matching frustum recess at the back to form circular window with a diameter of about 15 mm through which the OCT-device images the socket (Supplementary Fig. 3a–f). In total 24 conformers were produced, as each was mirrored to provide a conformer for left and right eye sockets.

Utilizing the SSM we predict a likely best fitting prosthesis shape based on the visible socket surface and the conformer's base shape. The OCT scan data of the eye socket is provided as 256 grey-scale bitmap images (Fig. 3d and Supplementary Fig. 4a), which are filtered and down-sampled to remove noise. The socket surface is extracted from this volumetric data (Fig. 3e) with a column-tracing based approach using the conformer window surface as reference points and for alignment (Supplementary Fig. 4f–h). After performing minor corrections, we encode the socket surface in a depth map $\mathbf{D}_S$ (Fig. 3f). While the OCT device captures an area of about 256 mm² only up to 177 mm² of the socket surface is visible because of the circular window; practically between 143 mm² and 174 mm² with a mean of 163(±9) mm² were extracted due to noise and alignment deviations. For comparison the projected area of posterior surface of the prostheses used to train the shape model is between 230 mm² and 736 mm² with an average of 420(±73) mm², thus on average 61% of the shapes posterior surface information is missing and must be predicted.

To find a fitting prosthesis shape we iteratively vary the shape parameter $\mathbf{x}$ to minimize an energy E($\mathbf{x}$) using the gradient based L-BFGS-B[35] algorithm (Fig. 3g and Supplementary Fig. 4k). The energy penalizes both the difference between the extracted surface depth map $\mathbf{D}_S$ and the projection of the back of the shape $S(\mathbf{x})$, and the weighted difference in shape space of $\mathbf{x}$ to a target shape $\mathbf{x}_t$. The parameter $\mathbf{x}_t = (1 − \alpha)\mathbf{x}_\mathbf{m} + \alpha\mathbf{x}_c$ is interpolated between the mean shape $\mathbf{x}_\mathbf{m} = \mathbf{0}$

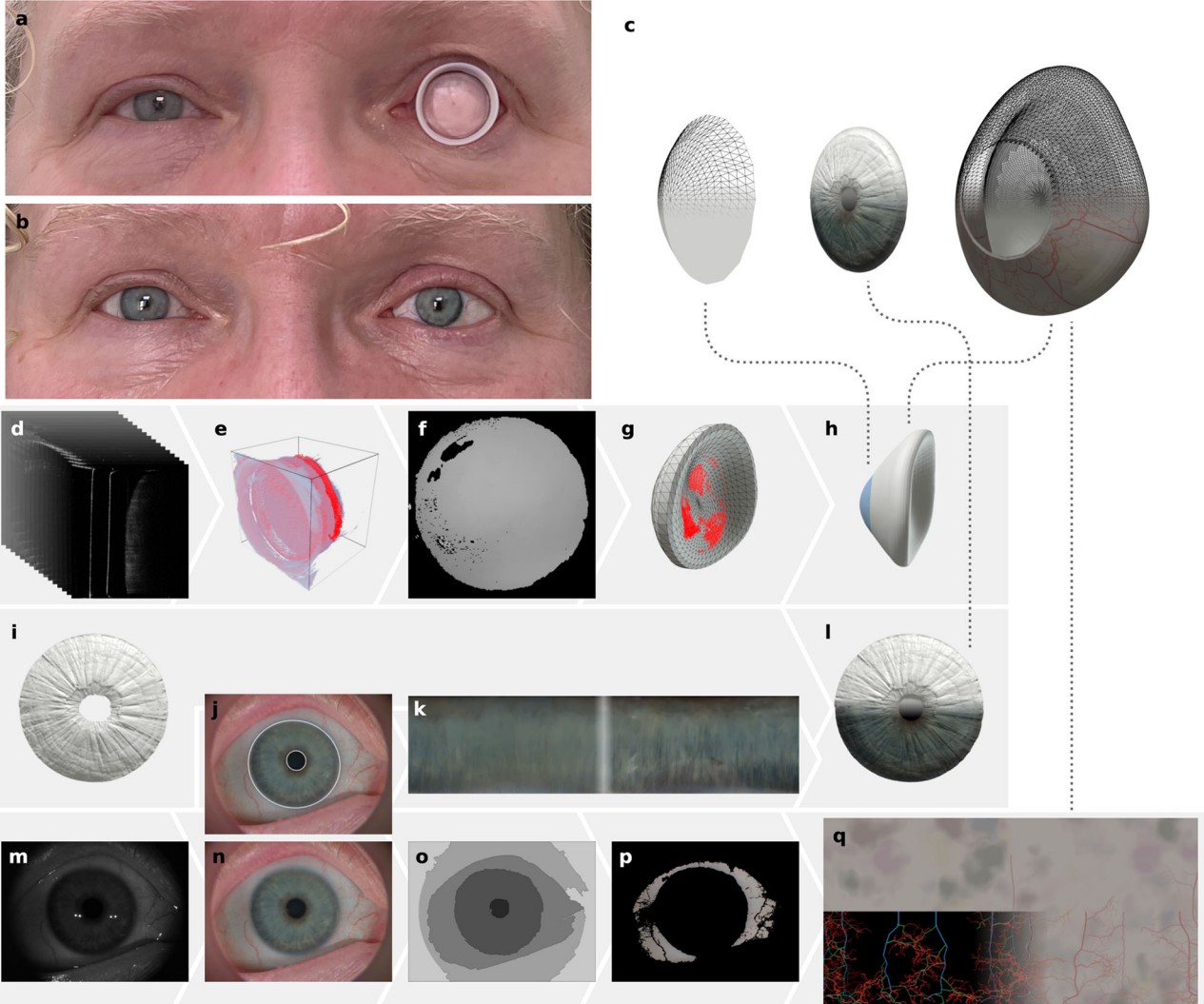

**Fig. 3 | Overview of data-driven design software process for the creation of the digital prosthesis model. a** Patient 5 with conformer in the eye socket. **b** Patient 5 with the 3D printed prosthesis of the left eye (right in the image). **c** Exploded 3D model of the prosthesis. **d**–**h** Shape prediction: OCT images of the eye socket (**d**), filtered volumetric data with extracted socket surface (**e**), converted to depth map for fitting (**f**), shape fitted to socket surface (**g**), smoothed and post-processed geometry (**h**). **i**–**l** Iris creation: Iris geometry provided by the OCT (**i**), iris and pupil boundary detection (**j**), unwrapped and contrast enhanced iris texture (**k**), normalized and UV-mapped iris geometry (**l**). **m**–**q** Colour image and sclera texture: Raw colour image (**m**), colour characterized, denoised and cleaned image (**n**), watershed segmentation (**o**), removed veins to extract sclera colours (**p**), rendering of staining texture and procedurally grown veining network (**q**, left), rendering of final sclera texture (**q**, right).

and the conformer base shape $\mathbf{x}_c$ of the used conformer, for each patient we predict three shapes using different target shapes using a bias $\alpha \in \{0.0, 0.5, 0.9\}$. In case any of the shapes is statistically unlikely our method assumes an alignment error and enables two degrees of freedom in the alignment: translation in the depth z and rotation around the vertical y-axis. The result is a shape $S(\mathbf{x})$ with the posterior surface matching the anophthalmic socket surface, and the anterior surface and transition fornix area predicted by the SSM. The choice of the conformer allows the ocularist to influence the prediction, to produce for example very thick prostheses for deep eye sockets.

It is also possible to determine the shape using the alignment and correspondence procedure for a 3D scan of an existing prosthesis or shape. This can be useful if a patient already has a prosthesis that fits well, if the ocularist made significant adjustments to the prostheses that affect the colour or appearance, or if a complex socket cannot be captured with the OCT device and requires that the ocularist manually makes a shape.

The determined prosthesis shapes are reshaped on the front; the cornea is replaced by the mean shape and rescaled to match the iris diameter, while maintaining smooth transitions into the shape (Fig. 3h). Since it is easier for the ocularist to remove material in the adjustment than to add material, which requires a reprint, we apply a locally varying scaling, that increases the size of the prosthesis by 5% without changing the size of the cornea, and add a locally varying clear coating via a displacement texture, that creates a thicker transparent shell on areas below the eye lids and the edges (Supplementary Fig. 4l, n). The geometry is smoothed using a curvature-based surface subdivision scheme (Supplementary Fig. 4m).

## Appearance reproduction

The appearance information is synthesized from a colour characterized image of the patient's eye. We denoise the raw colour image of the OCT device's camera (Fig. 3m), after flat fielding the image is converted to the CIELAB colour space using a two-step colour characterization method. Specular highlights in the image, resulting from the point-like illumination of the LEDs, are detected in the lightness channel, and removed via a simple inpainting algorithm (Fig. 3n and Supplementary Fig. 5b–d).

To extract the iris colour data, we detect the limbus and pupil boundary with a modified, multi-scale Daugman algorithm[36] (Fig. 3j). We unwrap the iris area using cylindrical coordinates and bilinear sampling onto a 4096 × 1024 pixels texture and apply a row-sensitive contrast enhancement of the lightness channel (Fig. 3k) to compensate contrast loss caused by light transport in the printing materials.

For the sclera texture creation, the image is segmented in four regions, pupil, iris, sclera, and aperture and skin, using extracted seed points and application of the watershed algorithm (Fig. 3o). We remove veins from the segmented region using a hue and chroma sensitive filter (Fig. 3p), and extract from this refined segmentation a set of 9 sclera colours using the k-means algorithm. We use Perlin noise masks to render the staining texture using the extracted sclera colours, on this we render a procedurally generated veining network[37] (Fig. 3q). The veining network is parametrized by two parameters, the vein thickness and the branching ratio. Based on anatomically motivated seed points the network is iteratively grown in three layers, where the growth of each vein is random and based on parameters that influence depth, width, branching, and path of the vein. The growth parameters of each vein are derived from a set of 5 manually defined vein recipes per layer, the grown network is rendered on the sclera texture using 15 vein profiles selected by the vein's width and depth (Supplementary Fig. 5j–o). These vein profiles represent a cross-section and are sampled from colour characterized images of eyes and labelled with a width and depth.

## Digital prosthesis model and production

The mesh of the iris disk, as provided by the OCT software (Fig. 3i), is refined at the boundary to produce a defined border at the pupil and a smooth transition to the limbus, closed at the pupil, and integrated at the limbus plane into the generated prosthesis shape (Supplementary Fig. 4m). A black inner cylinder is placed below the pupil to make the pupil very dark even with slightly translucent materials (Supplementary Fig. 5u). The generated textures for sclera and iris are mapped on the corresponding geometries using simple mapping transformations. Computing the digital prosthesis model requires in total about 5 min for each patient, on a desktop computer with an Intel i7-9700K and 64 GB of RAM. The textured 3D models of the ocular protheses are stored as OBJ files and prepared for 3D printing using the universal 3D printer driver Cuttlefish (Fraunhofer IGD, Germany), the data is sliced using a specifically created ICC colour profile which was optimized to accurately reproduce typical iris and sclera colours.

The slices are loaded into the GrabCAD software using the VoxelPrint utility tool and printed on the Stratasys J750 (Stratasys Ltd., Israel) PolyJet multi-material 3D printer using VeroVivid materials and High Mix mode, with a printing time of 6 min per prosthesis for a full print tray. The printed ocular prostheses are removed from the print bed and cleaned to remove the support material. The prosthetics are placed into a parts tumbler, which contains ceramic chips and water, and tumbled such that the 3D printer striations are removed and a homogeneous surface is achieved. Printing and these post-processing steps were performed by a commercial printing service provider (FIT AG, Germany). Prosthetics are inspected after tumbling, hand polished to a high shine, and inspected to ensure no irritating surface imperfections are present (Fig. 2k). The prosthetics go through a cleaning cycle in a sonication bath and following a final inspection, the each is distributed to the fitting ocularist.

The biocompatibility of the finished ocular prosthetics was assessed according to the ISO 10993 series of standards, which involved analytical chemistry, in vitro and in vivo testing to determine the likelihood of the device producing toxic responses in patients. This assessment showed that the prints are toxicologically safe for use.

## Patient supply

From late November 2022 to mid April 2023, the described process was successfully used to design and manufacture ocular prostheses for the 10 herein reported patients of the MEH ocular prosthetic department (4 male and 6 females). IRB approval was obtained from the Medical Devices and New Technology Committee of Moorfields Eye Hospital and was registered by the Audit Department, number CA23/RE/960. The patients with the supplied prosthesis are shown in Fig. 4. During the supply visit the ocularist adjusted the prosthesis' shape where necessary to improve the fit and function for the patient, see Supplementary Fig. 6. Adjustments are usually necessary in a bespoke fitting as certain aspects, such as the lid position, are not considered by the software. Fine tuning of the palpebral fissure is conducted in consultation with the patient.

The ocularist graded the prostheses' shapes after inserting each into the patient's anophthalmic socket. The grade describes whether it is possible to adjust a shape and how severe these adjustments are, results in Fig. 5a show that for 8 of the 10 patients at least one prosthesis could be adjusted. The ocularist selected the preferred prosthesis and adjusted its shape to achieve the desired facial and lid cosmesis. For Patient 6 all shapes were rated as excellent as no adjustments were necessary. For four patients at least one prosthesis shape was very good and had to be adjusted only once, for three patients it was acceptable and the ocularist had to adjust the shape at least twice during the fitting. We observed that for these patients only shapes with $\alpha = 0.0$ or $\alpha = 0.5$ were supplied and that for 7 of 10 patients (Patient 1, 3, 5–8, 10) all shape variants were graded the same, even though each variant has a different shape and size (e.g., Supplementary Fig. 3h). This is supported by the observation that, despite shape deviations, both a manually crafted prosthesis and a modified 3D-printed prosthesis can fit the same patient very well. Since there is not a definite best shape it may be important to give the ocularist a set of shapes to choose from.

For Patients 1 and 3 the prostheses' shapes were unacceptable and could not be supplied, both patients do not have an orbital implant and in both cases the produced prostheses were too big; additionally, there was no matching conformer for Patient 3 resulting in a divergent angle of gaze. For these two patients a 3D scan of a prosthesis, using the same setup and procedure as for the shape model shapes, was used to create, and print a digital artificial eye. For Patient 1 a 3D-printed prosthesis was modified, by removing material and adding wax, and scanned; for Patient 3 the patient's existing handmade prosthesis was scanned.

Since for all patients and all shape variants the predicted shape matches the extracted socket surface very closely, even for Patient 1 and 3 (Supplementary Figs. 3h, 6k, s), we believe that alignment errors or unsuitable conformers (that deform the eye socket during the scan) are the determining factor of the shape grade. This may be related to the observation that different shape variants are graded the same, since each fits the same extracted socket surface closely. For the unacceptable shapes the shapes of the hand-made prosthesis do not match the extracted socket surface (Supplementary Fig. 6m-p,u-x). We believe that for patients without an orbital implant the soft tissue presents a challenge to capture, as it can easily be displaced by the conformer.

When the desired fit has been achieved a cosmesis assessment was conducted in the same clinic room by the ocularist for each patient using a report, similar to a case report form, with 19 queries with four options (Excellent, Very good, Acceptable, Not Acceptable); the results are shown in Fig. 5b. Some of these strongly depend on the patient's anatomy, for example ptosis of the upper lid, laxity of the lower lid, upper and lower fornix depth can affect lid position that the shape of the prosthesis alone cannot correct and would require surgery.

One question rated the general facial appearance. Nine focus on the (adjusted) shape of the prosthesis. Upper and lower lid position are

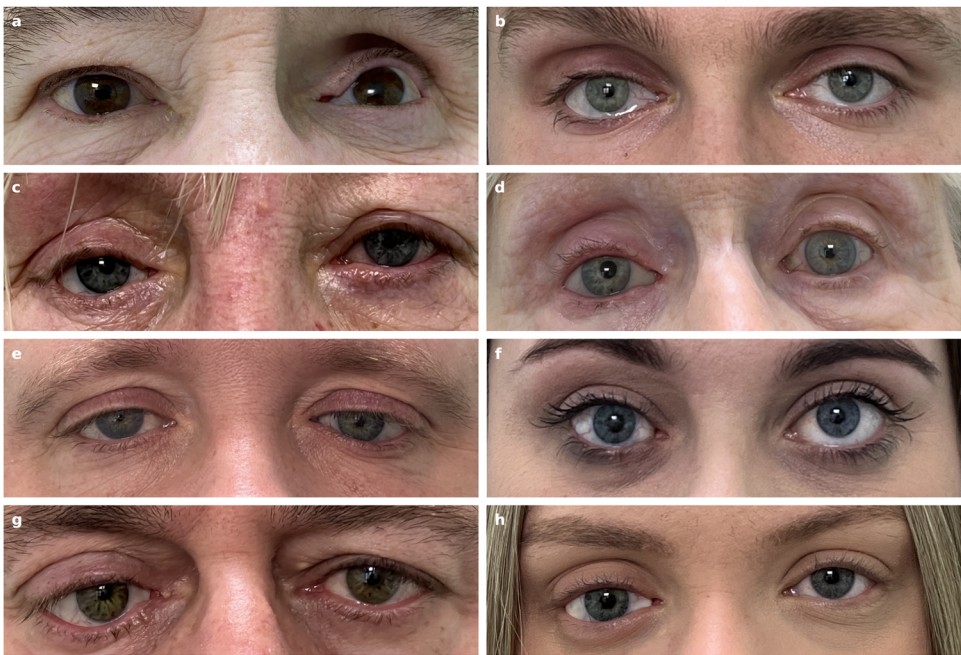

**Fig. 4 | Pictures of the automatically designed and 3D printed prostheses supplied to the patients. a–h** 8 of 10 Patients with their digitally designed, 3D printed prosthesis after adjustments. Patient 1 (**a**) and Patient 7 (**e**) have lost their left eye, with the prosthesis shown in the right of the image. Patient 2 (**b**), Patient 3 (**c**), Patient 6 (**d**), Patient 8 (**f**), Patient 9 (**g**), and Patient 10 (**h**) have lost their right eye, with the prosthesis shown in the left of the image. Patients 4 and 5 are shown in Figs. 2l and 3b.

compared to the fellow eye, but as mentioned the shape's influence can be limited. Fit and function is assessed to ensure that the prosthesis is secure in the socket and does not rotate in the socket or fall out if the eyelids are rubbed. Motility grades the comfort in all angles of gaze and how much the patient can move the prosthesis; it strongly depends on the orbital implant, which two patients do not have, and for this reason it is only acceptable for Patient 1. Gapping between the prosthesis and tissue is unavoidable for some patients, particularly if the prosthesis must be very small at the nasal side. Clicking, an effect where air between the prosthesis and the socket creates a noise, depends on the socket surface. Comfort is very subjective, it depends on personal preferences, condition of the tissue, and shape of previous prosthesis which requires the patient to get used to a different shape; for Patient 4 the comfort was acceptable because of that.

Further eight queries that focus on the appearance and colour match, also shown in Fig. 5b, were rated at least very good. Four rate the size and shape of both the iris and pupil, here one difficulty is that the pupil size varies with lighting conditions. Four also rate the colour match and the details of the iris and the sclera; for the sclera the veining and to some extend the colour depend on irritation of the eye. One question allowed the patient to rate their satisfaction with the prosthesis, 7 rated the prosthesis as excellent and 3 as very good, here the realism and colour match of the iris was the determining factor. Depending on the adjustments it took about 30 to 90 min, with an average of about 60 min, to try, adjust, polish, assess and supply the prosthesis; the assessment is not necessary in a clinical setting.

## Discussion

We report a process for the digital and automatic manufacture of ocular prosthetics that fit the patients' eye sockets and match their fellow eye's appearance, demonstrated by our preliminary clinical assessment. It is currently the subject of a clinical trial at MEH (NCT05093348, https://clinicaltrials.gov) with a different group of patients, which assesses the long-term impact and investigates the performance compared to traditionally manufactured prostheses; an ancillary study evaluates whether the process is suited for cosmetic

shell, these very thin prostheses are required by patients with defectively small eyes due to microphthalmia.

We believe there are significant benefits. First, using a colour calibrated printer, colour characterized images, and the AS-OCT data our method accurately replicates the colour and anatomy of the fellow eye, in particular the colour, size, and structure of the iris, but also the appearance of the sclera. Second, this process requires less manual labour time compared to a traditional process, allowing an ocularist at the MEH to produce about five times as many prostheses. It requires a certain volume to be cost effective, but could translate into cost savings once scaled up. Conceivably prosthetics could be supplied remotely, especially in areas where such services are not readily available. Third, compared to the manual manufacture the output is very consistent, whereas the skills of ocularists can vary, and reproducible, allowing the uncomplicated supply of spares or replacements of lost prostheses. Fourth, a possibly uncomfortable alginate impression is unnecessary as the socket shape is scanned optically, using OCT to image the eye socket avoids ionizing radiation. Fifth, a digital workflow allows for continuous improvements that are available to all patients without additional training of the ocularist. Using more data and the ocularists' and patients' feedback the software can be improved and refined, to better determine shapes or replicate certain features of the eye. These changes could also allow to supply patients that were previously not eligible, for example children. Advances in the materials or printers allow improved reproduction of colours and details; better AS-OCT devices can cover a deeper volume or provide higher-quality colour images.

However there are some limitations, the prostheses usually require a final adjustment of the shape and fitting by the ocularist, making the output modified custom ocular prostheses; while there are some manual steps in the process left most of the manufacturing is automated. Very complex sockets cannot be captured by the OCT device or the shape model and still need the manufacturing craft of the ocularist for the shape. Currently about 80% of the patients that require an ocular prosthesis are eligible for the described process; patients with certain eye conditions are not suitable, for example the

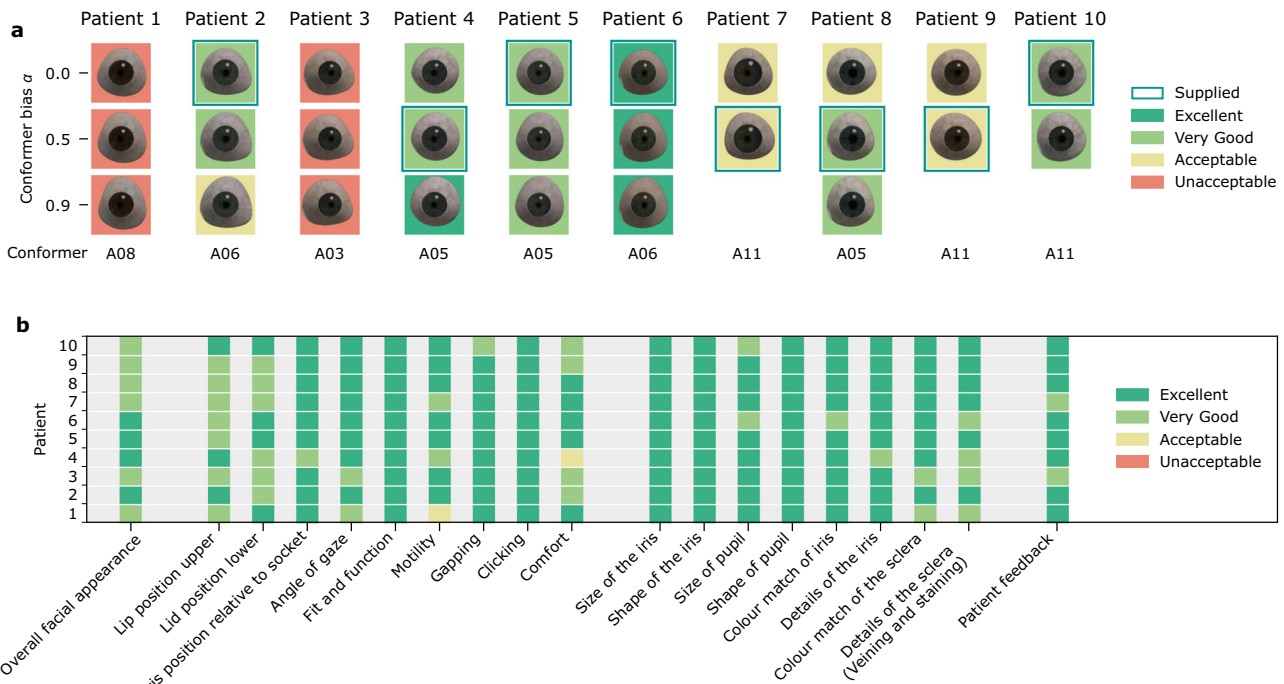

**Fig. 5 | Evaluation of shape and cosmesis of the produced prostheses for the 10 patients. a** Prostheses' shape variants, with different conformer biases $\alpha$ towards the conformer base shape, were graded by the ocularist; the conformers are shown in Extended Data Fig. 4. For three patients the shape for $\alpha = 0.9$ was larger than the allowed size and not produced; for Patient 1 and 3 the shapes were unacceptable and could not be supplied; Patient 4 found the shape for $\alpha = 0.5$ more comfortable than the better graded shape for $\alpha = 0.9$. **b** Results of the cosmesis assessment, which was done for all patients after the ocularist made the adjustments; for almost all queries of the report the result was very good or excellent, motility was acceptable for Patient 1 because this patient does not have and orbital implant, comfort was acceptable for Patient 4 because the patient needs to get used to the new prosthesis shape.

required scan data cannot be obtained from patients who suffer from nystagmus or strabismus.

We envision that our findings motivate the research and development of data-driven design tools and multi-material 3D printing for other prosthesis types such as dental restorations or facial prostheses. It could even be possible to couple this development with more traditional prostheses and for example produce covers for prostheses that accurately match the patients' appearance.

## Methods
### OCT device
All patient data was captured with a modified prototype Casia 2 (Tomey Corporation, Japan) AS-OCT device which uses a swept laser source with a wavelength of 1310 nm and less than 6 mW power to scan a $16 \times 14$ mm$^2$ slice at a resolution of $2145 \times 1877$ pixels (width and depth), 256 slices are combined to capture a $16 \times 16 \times 14$ mm$^3$ raster scan volume.

The device was modified by the manufacturer with a AR0135CS CMOS (ON Semiconductor Corporation, U.S.) colour camera with $1280 \times 1024$ pixels in Bayer GR pattern with a 12 bit depth, and a specific light consisting of YUJILEDS (Yuji International Co., Ltd., China) full spectrum high CRI 98 D50 LED light with 45°/0° measurement geometry. The light source has been selected to have a spectral power distribution that spectrally nearly matches the CIE D50 daylight illuminant used for the printer characterization to minimize metamer mismatch uncertainties in the workflow. The measurement geometry has been selected to match the measurement geometry of the Barbieri Spectro LFP qb (BARBIERI electronic snc/OHG, Italy) spectrophotometer used for the printer characterization.

Software modifications include the export of the raw colour sensor image data, iris surface geometry, and of 256 raw raster scan images for each patient scan.

### Shape 3D scan and alignment
We align the ocular prosthesis shapes given by 3D scans of existing prosthesis with the following procedure. First the prosthesis is marked up with a fine permanent marker pen (Supplementary Fig. 1a), the markings consist of a ring at the limbus and a crosshair indicating the iris centre point and directions (superior, nasal, inferior, and temporal). This marking describes four intersection points between the ring and the crosshair. The prosthesis is scanned using a MEDIT T500 Dental 3D scanner (MEDIT Corp, South Korea), shown in Supplementary Fig. 1b, c, with the setting texture marking turned on and mesh exported as an OBJ file. The scan mesh data is loaded into MeshLab[38], the four intersection points ($\mathbf{p}_S$, $\mathbf{p}_N$, $\mathbf{p}_I$, $\mathbf{p}_T$) are annotated manually using its PickPoints feature.

To align and process the prostheses shapes we define a coordinate system such that the iris lies in the x-y-plane, also denoted as the iris plane, with the iris centre at the origin. The positive x direction points nasal (to the nose) and opposite negative x direction temporal (to the ear), the positive y direction points superior (to the top) and negative y inferior (to the bottom), and the positive z direction points towards the cornea dome and negative z inside the eyeball or eye socket.

For the alignment (Supplementary Algorithm 1) we apply a translation by the vector $\bar{\mathbf{p}} = -(\mathbf{p}_S + \mathbf{p}_N + \mathbf{p}_I + \mathbf{p}_T)/4$ and a rotation $\mathbf{R} \in \mathbb{R}^{3 \times 3}$, which is the least squares solution of

$$\mathbf{PR} = \left[ \begin{pmatrix} 1 \\ 0 \\ 0 \end{pmatrix} \begin{pmatrix} 0 \\ 1 \\ 0 \end{pmatrix} \begin{pmatrix} -1 \\ 0 \\ 0 \end{pmatrix} \begin{pmatrix} 0 \\ -1 \\ 0 \end{pmatrix} \right]^T \tag{1}$$

where $\mathbf{P} = \left[ (\mathbf{p}_N - \bar{\mathbf{p}})(\mathbf{p}_S - \bar{\mathbf{p}})(\mathbf{p}_I - \bar{\mathbf{p}})(\mathbf{p}_T - \bar{\mathbf{p}}) \right]^T \in \mathbb{R}^{4 \times 3}$.

This transforms the geometry and intersection points to lie in the iris such that the centre point of the limbus ring is at the origin, the limbus ring lies in the x-y-plane, and the intersection points lie on the

respective x-axis or y-axis (Supplementary Fig. 1d). Meshes of left eye prostheses are mirrored along the y-z-plane so that all prosthesis shapes fit a right eye socket.

## Correspondence generation

We compute corresponding landmarks on the mesh surface via depth maps (Supplementary Algorithm 2), using the fact that the maximum number of intersections between an aligned prosthesis shape' surface and any line parallel to the z-axis is two. The depth maps are given by the orthogonal projection of the surface along the z-Axis for front and back view (Supplementary Fig. 1f, j). To determine the corresponding points, we perform ray marching to compute lines radially from the origin to the edge of the prosthesis projection, for 48 directions at incremental angles of 7.5°. On these lines we distribute points at distinct fractions (shown in Supplementary Fig. 1g, k), points placed at the same fraction form a ring. We skip the placement at certain fractions for lines at distinct angles to achieve a more homogeneous density, and for the front we select the factions such that the limbus is always marked by the third ring. The result is a point cloud $\mathcal{P} \in \mathbb{R}^{838 \times 3}$ with 838 corresponding vertices (189 vertices on the front and 649 on the back of the prosthesis shape); together with a set of 1672 triangle faces $\mathcal{F}$ it defines the shape's mesh representation $\mathcal{S} \in \Theta$ (Supplementary Fig. 1h, l), with $\Theta$ being the space of the prosthesis shape vertices and faces.

## Shape model generation

A selection of 173 manually produced prostheses sourced from a stock at MEH were scanned, marked, aligned, and the corresponding landmarks computed as described above. The size of the axis-aligned bounding box (width, height, and depth) of the shapes ranges from $(18.5 \times 15.9 \times 9.7)$ mm$^3$ to $(28.6 \times 32.3 \times 22.2)$ mm$^3$ with a mean of $(24.2 \times 22.5 \times 15.5)$ mm$^3$. We did not perform Generalized Procrustes Analysis[39] for two reasons. First because the correct rotation is already given and the size is normalized implicitly by the limbus size that was very similar for these shapes. Secondly and more importantly the size of the prosthesis is related to the shape, meaning that some shapes are more likely for certain eye socket volume sizes.

For the statistical shape model (SSM) we perform a standard principal component analysis (PCA) using singular value decomposition (SVD) on a matrix composed from the flattened vertices positions (Supplementary Algorithm 3). Reshaping this into the vertex list yields the shape model

$$S(\mathbf{x}) : \mathbb{R}^k \mapsto \Theta = \mathcal{S}_m + \sum x_i \sigma_i \mathcal{C}_i \qquad (2)$$

with shape vector $\mathbf{x} \in \mathbb{X} = \mathbb{R}^k, k \in \mathbb{N}$, mean shape $\mathcal{S}_m$, and principal components or modes $\mathcal{C}_i$ being scaled by the standard deviation $\sigma_i$ given by the singular values $\lambda_i = \sigma_i^2$, which allows a uniform treatment of the modes. Here $\Theta$ is the space of eye prostheses' meshes with 838 corresponding vertices and 1672 faces, since the set of faces $\mathcal{F}$ is identical for all shapes we omit it in the equations for simplicity. We define $S^{-1}(\mathcal{S})$ as the inverse transformation that maps a shape $\mathcal{S} \in \Theta$ into the shape space $\mathbb{X}$, note that $S^{-1}(S(\mathbf{x})) = \mathbf{x}$ but generally $S(S^{-1}(\mathcal{S})) \neq \mathcal{S}$.

Since the shapes are not normalized by bounding box size, the first modes encode majorly changes of the prostheses size. Since the shape of the prostheses correlates with the size and bigger prosthesis tend to have different shapes than smaller ones, this is desired for the shape predictions sake. An overview of the shape variations of $S(\mathbf{x})$ for the first five modes is given in Supplementary Fig. 2f.

We select the number of components so that they explain 98% of the shape variance; the training then yields a SSM with $k = 17$ components. The shape model reproduces the training sample shapes with a mean corresponding vertex distance of 0.27 (±0.06) mm; the maximal vertex distance per sample is on average 1.17 (±0.36) mm, and at most 2.59 mm (Supplementary Fig. 2d, e). Note that the Hausdorff distance between the vertices is smaller and still an upper boundary for the

Hausdorff distance of the shape surface. The generalization, which describes the ability of the SSM to represent unseen data, was computed by training the SSM with each example excluded from the training set and then recreated. As shown in Supplementary Fig. 2b it achieves a mean vertex distance of 0.31 (±0.09) mm. The specificity, which describes the plausibility of the SSM's shape space, was computed by randomly sampling 10000 shape parameters from a normal distribution and measuring the difference to the most similar example from the training set. This shows a mean vertex distance of 1.10 (±0.23) mm (Supplementary Fig. 2c).

## Conformer generation and production

A selection of 12 prosthesis shapes, that represent the most common ocular prosthesis shapes, were scanned, aligned (Supplementary Fig. 3a, d), and processed yielding the shapes $\mathcal{S}_c, c \in \{1, \ldots, 12\}$; these shapes have an axis-aligned bounding box size of between $21.2 \times 17.8 \times 10.4$ mm$^3$ and $26.7 \times 27.4 \times 19.7$ mm$^3$. A shape vector $\mathbf{x}_c = S^{-1}(\mathcal{S}_c)$ in the shape space is computed for each. The shapes $S(\mathbf{x}_c)$, also called the conformers' base shapes (Supplementary Fig. 3b, e), were modified by cutting an anterior window surface parallel to the x-y-plane or iris plane with an offset of 1.8 mm and carving a conical frustum with its top surface parallel to form a circular window with a diameter of about 15 mm and a thickness of 1.5 mm (Supplementary Fig. 3c, f). The anterior surface is offset below the iris plane to effectively increasing the depth of the raster scan. Each conformer was 3D printed twice, once for left and right eyes, with the Stratasys J750, using the VeroClear material which is transparent to the OCT laser, and polished. A ring lid lift is added to the front to keep the eye lids open. Information about the shape vector $\mathbf{x}_c$ and window geometry, specifically window thickness and offset to iris plane, are stored in a library with the conformer identifier and retrieved by the software during processing. The 12 conformers are shown in Supplementary Fig. 3c, f–i. Some conformers (A01, A02, A04, A07, A09, A10, and A12) were not selected for any of the 10 patients while others such as A05 and A11 were selected three times (Fig. 5a).

## Socket surface extraction

The OCT raster scan data of the eye socket is exported as 256 grayscale bitmap slices, each slice image $\mathbf{B}_{Input}$ has 2145 × 1877 pixels with 8-bit depth and lies in the x-z-plane (Supplementary Fig. 4a). For $\mathbf{B}_{Input}$ a threshold $b_{th} = 25$ is applied with

$$\mathbf{B}'_{Input} = (\max(\mathbf{B}_{Input}, b_{th}) - b_{th})/(255 - b_{th}) \qquad (3)$$

To remove noise the image $\mathbf{B}'_{Input}$ is median filtered and downscaled to $\mathbf{B}_{Median}$ using repeated application of median filters, Median Pooling and Max Pooling (Supplementary Algorithm 4) resulting in the image $\mathbf{B}_{Median}$ of size 536 × 469 pixels (Supplementary Fig. 4b). Since the conformer window surface becomes faint in $\mathbf{B}_{Median}$ we compute a gradient based edge map $\mathbf{B}_{Edge}$ (Supplementary Fig. 4d) using another downscaled image $\mathbf{B}_{Max}$ which is computed from $\mathbf{B}'_{Input}$ using simple 4 × 4 Max Pooling. Then the binary image $\mathbf{B}_{Edge}$ is computed with

$$\mathbf{G}_x = \frac{\partial^2}{\partial x^2} \left( \mathbf{B}_{Max} \odot 1_{\{x \in \mathbf{B}_{Median} : x > 0\}} (\mathbf{B}_{Median}) \right) \qquad (4)$$

$$\mathbf{B}_{Edge} = \mathbf{G}_x < -0.05 \qquad (5)$$

using the Sobel Operator with a kernel size of 5 for the 2$^{nd}$ derivative, the indicator function $1_A(\mathbf{B})$ and Hadamard product $\odot$ effectively mask $\mathbf{B}_{Max}$ with non-zero pixels of $\mathbf{B}_{Median}$. Note that the image's x-direction coincides with the volumetric z-axis or depth. This

procedure is performed for all 256 slices and the values of $\mathbf{B}_{\text{Median}}$ and $\mathbf{B}_{\text{Edge}}$ are stored in tensors $V_{\text{Median}}$ and $V_{\text{Edge}}$ of size $536 \times 256 \times 469$ voxels (width, height, and depth) describing the socket volume (Supplementary Fig. 4c, e).

To determine the anterior and posterior conformer window surface from $V_{\text{Edge}}$ we use pixel-wise column tracing in the z-direction or depth, denoting $z$ the depth within a column and $v_{\text{E}}(z)$ the corresponding value (Supplementary Algorithm 5). Starting from a depth $z_0 = 0$ we trace each column to find the first position $z_1$ with $v_{\text{E}}(z_1) > 0$, each depth $z_1$ is encoded in a $536 \times 256$ pixels depth map $\mathbf{D}_A$. We initialize a binary mask $\mathbf{M}_A = \mathbf{D}_A < (\text{median}(\mathbf{D}_A) + 0.75\tau)$ of the same size, which marks entries of the depth map that belong to the anterior window; $\tau$ is the conformer thickness in voxels. We determine the anterior surface via iterative refinement of $\mathbf{M}_A$: We perform a least-squares fit of a plane $P_A(x,y)$ to the pixel values of $\mathbf{D}_A$ in the mask $\mathbf{M}_A$, and set $\mathbf{M}_A(x,y) = (|\mathbf{D}_A(x,y) - P_A(x,y)| < 0.5\tau)$. This refinement iteration is repeated 8 times, then the entries of $\mathbf{D}_A$ marked in $\mathbf{M}_A$ represent the anterior window surface (Supplementary Fig. 4f).

The posterior window surface depth map $\mathbf{D}_B$ is computed similarly, here the column tracing is started with depths $\mathbf{D}_B = (\mathbf{D}_A + 0.5\tau)$ and $\mathbf{M}_B$ is initialized with $\mathbf{M}_A$, the refinement iteration is performed 4 times. Median filter and morphological opening are used to close gaps in the binary mask $\mathbf{M}_B$ that defines entries of $\mathbf{D}_B$ belonging to the posterior window surface (Supplementary Fig. 4g).

We perform column tracing of columns in $V_{\text{Median}}$, with values $v_M(z)$, to extract the anophthalmic socket surface below the conformer window, starting for each column in the mask $\mathbf{M}_B$ with $\mathbf{D}_B + 10$ as $z_0$. However due to noise the first encountered non-zero entry of the column may not be the socket surface. Instead, we first find the depth $z_1 = \text{argmax}_{z > z_0} v_M(z)$ of each column with the highest value. We find the position $z_2 = \text{argmax}_{z \in \{z_0 + 1, ..., z_1\}} (v_M(z) - v_M(z_0))/(z - z_0)$ such that the position $z_2$ represents a line with the steepest slope to $z_0$. The socket surface $z_3$ is given by the position $z_3 = \max\{z | z < z_2 \wedge v_M(z) = 0\}$ as the previous column position with a value of zero. In the resulting depth map $\mathbf{D}_S$ we use a Median Filter to detect and remove outliers, then convert it into a point cloud $P_S$ to apply three corrections (Supplementary Algorithm 6, Supplementary Fig. 4h, i).

First, we correct changes of the optical path due to the tilt of the conformer window using Snell's law with a refractive index of 1.5 (for the conformer's VeroClear material) and then apply an empiric correction of the depth by $-0.35d$, depending on the effective conformer thickness $d$ in mm (surface signals appear closer if the light travels through the conformer). Second, we align the surface with a transformation along the z-axis such that the anterior conformer window plane $P_A$ intersects the origin and correct the shift due to the offset of the conformer window to the iris plane. Third, we correct for the angle of gaze being different from the optical axis assuming an angle of 6°.

The corrected point cloud $P_S$ is converted to a $256 \times 256$ pixels depth map $\mathbf{D}_S$ of the socket surface (Supplementary Fig. 4j), a binary mask $\mathbf{M}_S$ indicates pixels that belong to the socket surface. If the extracted mask covers an area less than $32 \text{ mm}^2$ the process is aborted, the smallest observed was $142.7 \text{ mm}^2$ for Patient 10.

## Shape prediction

To find the best fit prosthesis shape $S(\mathbf{x}_f)$ for a given socket surface depth map $\mathbf{D}_S$ and a target shape $\mathbf{x}_t$ we minimize an energy $E(\mathbf{x}) = E_{\text{dist}}(\mathbf{x}) + E_{\text{ref}}(\mathbf{x})$ by variation of $\mathbf{x}$ (Supplementary Algorithm 7).

The term

$$E_{\text{dist}}(\mathbf{x}) = \frac{w_{\text{dist}}}{\text{su}(\mathbf{M}_S)} \text{su}(\mathbf{M}_S \odot (Z_z(S(\mathbf{x})) - \mathbf{D}_S)^{\odot 2}) \tag{6}$$

penalizes the difference between the back of the prosthesis and the visible socket surface, with $\text{su}(\mathbf{A})$ the grand sum of all elements of $\mathbf{A}$, $\odot$ the Hadamard product, and $\mathbf{A}^{\odot 2}$ being the Hadamard power of 2. $Z_z$ :

$\Theta \to \mathbb{R}^{256 \times 256}$ maps the back of the prosthesis shape $S(\mathbf{x})$ to a depth map via orthographic z-projection, which is compared with the socket surface depth map $\mathbf{D}_S$ (see Supplementary Fig. 4k). The difference is element-wise squared, masked with $\mathbf{M}_S$, summed up, normalized with the number of entries in $\mathbf{M}_S$ and weighted with $w_{\text{dist}}$.

The term $E_{\text{ref}}(\mathbf{x}) = ||\mathbf{w}_{\text{ref}}(\mathbf{x} - \mathbf{x}_t)||_2$ penalizes the weighted difference between the shape $\mathbf{x}$ and the target shape $\mathbf{x}_t$ in the shape space. The weights were empirically chosen as $w_{\text{dist}} = 10000$, $\mathbf{w}_{\text{ref}} = (2048, 1024, 512, 256, 128, 64, 32, 16, 16, 16, ...)^T$ is weighted similar to the shape model variances of the corresponding modes.

Starting with $\mathbf{x} = \mathbf{x}_t$ we minimize $E(\mathbf{x})$ using the L-BFGS-B algorithm with the solution space constrained to a hypercube of $\pm 3$, corresponding to $\pm 3$ standard deviations. We enable the modes sequentially in steps of 3 during the fitting process for a coarse to fine fitting, an example of the fitting process and its intermediate results is shown in Supplementary Fig. 4k.

If the solution $\mathbf{x}_f$ contains a weight at the hypercube edge $\pm 3$, indicating a very unlikely shape with weights corresponding to three standard deviations, we assume alignment errors and allow additional degrees of freedom. The additional translation $\mathbf{t}_z(z) = (0, 0, z)^T$ in mm along the z-Axis and rotation $\mathbf{R}_y(\theta)$ of $\theta$ radians around y-Axis, both transformations applied to each vertex of $S(\mathbf{x})$, is penalized in the energy

$$E(\mathbf{x}, \theta, z) = E_{\text{dist}}(\mathbf{x}, \theta, z) + E_{\text{ref}}(\mathbf{x}) + w_{\text{ext}}\left(\theta^2 + z^2\right) \tag{7}$$

$$E_{\text{dist}}(\mathbf{x}, \theta, z) = \frac{w_{\text{dist}}}{\text{su}(\mathbf{M}_S)} \text{su}(\mathbf{M}_S \odot (Z_z(\mathbf{R}_y(\theta)S(\mathbf{x}) + \mathbf{t}_z(z)) - \mathbf{D}_S)^{\odot 2}) \tag{8}$$

where the rotation $\theta$ is limited to $\pm 30°$, and translation $z$ is limited to $\pm 2.5$ mm, and $w_{\text{ext}} = 10000$.

We determine three prosthesis shapes using three target shapes derived from the mean shape $\mathbf{x}_m = \mathbf{0}$ and conformer shape $\mathbf{x}_c$ by linear interpolation $\mathbf{x}_t = (1 - \alpha)\mathbf{x}_m + \alpha\mathbf{x}_c = \alpha\mathbf{x}_c, \alpha \in \{0.0, 0.5, 0.9\}$. The influence of the target shape $\mathbf{x}_t$ and bias $\alpha$ on the predicted shape $\mathbf{x}_f$ is visualized in Supplementary Fig. 3h.

## Shape reconstruction

To create a prosthesis based on an already existing prosthesis shape a marked 3D scan is used, the alignment and correspondence procedure provides the shape $\mathcal{S}_{\text{rec}}$ being directly given by the corresponding points. It is processed further identical to $S(\mathbf{x}_f)$, except that it is not locally scaled up by 5% and the clear coating thickness has a uniform thickness of 0.2 mm. Note that we directly use $\mathcal{S}_{\text{rec}}$ instead of its shape representation $S(S^{-1}(\mathcal{S}_{\text{rec}}))$, because of $S(S^{-1}(\mathcal{S})) \neq \mathcal{S}$ this allows to replicate even shapes that are not explained well by the SSM.

## Shape post-processing

The determined shape $S(\mathbf{x}_f)$ (or reconstructed shape $\mathcal{S}_{\text{rec}}$) requires some adjustments of the anterior surface, specifically it is necessary to adjust the cornea to match the iris diameter (Supplementary Algorithm 8). We normalize the shape's cornea region using the cornea of the mean shape, scale the cornea such that the apex lies 2.5 mm from the origin (this value was chosen empirically based on test prints), and scale the x- and y-components such that the limbus diameter matches the iris diameter; these modifications only affect the anterior surface of the prosthesis shape and transition smoothly into the edges. The shape's surface is smoothed two times via a subdivision scheme similar to Loop subdivision[40], that uses the vertex normal to place the new vertices outside the shape such that the volume increases (instead of decreases), shown in Supplementary Fig. 4m.

For regulatory compliance safety reasons shapes with a bounding box size larger than $30 \times 29 \times 20 \text{ mm}^3$ (width, height, and depth) are discarded; this happened for Patient 7, 9, and 10, here shapes based on

the conformer A11, which has a bounding box size of $24.4 \times 26.2 \times 18.6$ mm$^3$ and biases the shape prediction towards deeper shapes, were too large.

## End-to-end colour characterization

Colour reproduction of the whole end-to-end workflow is optimized for CIE D50 illuminant and CIE 2°-standard observer and a 45°/0° viewing geometry. Camera RGB pixel values are mapped to CIELAB values by two successive transformations (Supplementary Algorithm 9). A linear transformation $f_A(\mathbf{c}) = \mathbf{Ac}$ maps the camera RGB values $\mathbf{c} \in [0,1]^3$ to CIEXYZ values, where $\mathbf{A}$ is a $3 \times 3$ matrix. The resulting CIEXYZ values are transformed to CIELAB values which are then corrected by a root-polynomial transformation:

$$\left(L^*, a^*, b^*\right) = \Gamma\left(f_A(\mathbf{c})\right) \tag{9}$$

$$g_B\left(L^*, a^*, b^*\right) = \mathbf{B}\left(L^*, a^*, b^*, \sqrt{L^* a^*}, \sqrt{L^* b^*}, \sqrt{a^* b^*}, 1\right)^T \tag{10}$$

where $\Gamma : \text{CIEXYZ} \rightarrow \text{CIELAB}$ is computed assuming CIE D50 illuminant and the CIE 2° standard observer and $\mathbf{B}$ is a $3 \times 7$ matrix.

Transformations $f_A$ and $g_B$ are fitted to a set of captured and measured colour patches, consisting of 24 Colour Checker, 24 iris as well as 15 sclera colours. The latter two sets are selected aiming to cover the most common iris and sclera colours found in the population. For this, samples were prepared by an experienced ocularist with the colour inks and materials used to create manual ocular prostheses. These samples were measured, and their colour added to the colour patches. Measurements are conducted with the Barbieri Spectro LFP qb (BARBIERI electronic SNC, Italy) spectrophotometer. The resulting colour characterization has a mean CIEDE2000[41] difference of 1.95 ($\pm$1.67) and maximum difference of 7.54 for the colours of these colour patches, the accuracy of the characterization is limited by the camera spectral sensitivities.

The printer is characterized using the Robust Plausible Deep Learning (RPDL) optical printer model[42,43] trained on 4172 samples. Mean prediction accuracy measured on 300 randomly selected tonal test samples is CIEDE2000 = 1.1 and the 90th percentile accuracy is CIEDE2000 = 1.9. A perceptually optimized Colour Lookup Table (CLUT) is generated for the backward (CIELAB to 3D-printer tonal value space CMYKWClear) transformation[44]. To resolve colorimetric redundancies, the backward transform uses maximum Grey Component Replacement (GCR) to maximize colour constancy of the prints.

## Colour image processing

Image acquisitions are conducted in a controlled and equal lighting conditions. For this, all pictures for the characterization and of patients' companion eyes are taken in a dark environment to avoid any variations of stray light impacting colour accuracy.

The following procedure is conducted to process the raw image $\mathbf{I}_{raw}$ of the OCT's colour camera (Supplementary Algorithm 10, Supplementary Fig. 5a): Demosaicing of the raw camera image $\mathbf{I}_{raw}$ to obtain Image $\mathbf{I}$ using a standard edge-aware Debayer method of the OpenCV library[45]; denoising of $\mathbf{I}$ with non-local means[46] using a high filtering parameter of 15; a Pixel-wise thermal noise correction $\mathbf{I}' = \mathbf{I} - \bar{b}$ where $\bar{b}$ is the mean pixel value of an image $\mathbf{I}_{raw}$ taken under no light; flat fielding with $\mathbf{I}'' = \bar{w}(\mathbf{I}' \oslash \mathbf{W})$ where $\mathbf{W}$ is the thermal noise corrected image of a uniform white patch with mean pixel value $\bar{w}$ (and $\oslash$ is the Hadamard division); colour characterization of $\mathbf{I}''$ and denoising with non-local means using a low filtering parameter of 1.5 yields the image $I_{col}$ in CIELAB colour space (Supplementary Fig. 5b). To remove specular highlights in $I_{col}$ resulting from the LED-based point light sources, we identify a mask $\mathbf{M}_H$ of pixels in the raw image $\mathbf{I}_{raw}$ that are oversaturated or where the lightness differs more than $\Delta_{L^*} = 10$ from

the neighbourhood's median lightness. This mask $\mathbf{M}_H$ is dilated and we use a simple inpainting algorithm[47] to paint over the specular highlights, yielding $I_{clean}$ (Supplementary Fig. 5c).

## Iris mesh and image processing

The CASIA software directly provides a STL mesh of the iris surface (Fig. 3i), which is aligned to lie in the x-y-plane with the iris centre at the origin. We compute an average pupil diameter and normalize the pupil boundary to form a smooth circle; the pupil hole is then closed (Fig. 3l). To enhance the contrast between particular the iris and the sclera region we compute a greyscale image $I_{enh}$ by converting the image $I_{clean}$ to CIELCh and then computing the difference between the chroma and the lightness channel (Supplementary Fig. 5d). We compute a segmentation of the iris and pupil with multi-scale application of the Daugman[36] algorithm at four different scales on the image $I_{enh}$ to compute the centre and radius of both iris and pupil. To compute the elliptical iris boundary, we apply an additional step that deforms the circle to ellipses of constant area but different eccentricities and angles to maximize the difference between the sums of the intensities outside and inside the ellipsis (Supplementary Algorithm 11). The result is shown in Supplementary Fig. 5e, q.

Using this segmentation $\mathbf{M}_I$ of the iris and pupil region in the colour characterized image $I_{clean}$ we unwrap the iris to a texture $T_I$ with cylindrical coordinates (Supplementary Fig. 5r). To correct for contrast loss due to light transport within the printing materials, we enhance the contrast of the lightness channel in the unwrapped texture $T_I$ using the following procedure: Compute the mean lightness $L_0^*$ of the entire image and a row-wise lightness mean $L_r^*$ for each row $r$; stretch the differences of each lightness value at row $r$ and column $c$ via $L_{r,c}^* = 1.5\left(L_{r,c}^* - L_r^*\right) + L_r^*$; compute the mean lightness $L_1^*$ of the image and add the difference to $L_0^*$ to each pixel to ensure that the overall lightness remains unchanged (Supplementary Fig. 5s). We set the bottom of the image, that corresponds to the pupil, to pure black. In the last step we map the texture on the iris geometry (Supplementary Fig. 5t) and add a black cylinder behind the pupil (Supplementary Fig. 5u), this results in a darker pupil if the printer's black material exhibits light transport.

## Sclera segmentation

We compute $I_{median}$ from $I_{col}$ by iteratively applying median blurring 20 times with kernel sizes 5 and 3 (Supplementary Fig. 5f). For the segmentation we determine seed points in a central, horizontal cross-section of the image; for the pupil we search the darkest pixel and for the sclera brightest pixels left and right of that. Iris seed points are located by evaluating the derivative of the cross-section in the area between the sclera and pupil seed points; the extrema (of the changes in pixel brightness) correspond to the limbus and pupil boundary with the iris in between. Using these seed points, we compute a watershed segmentation[48] $\mathbf{M}_E$ of the image $I_{median}$ in pupil, iris, sclera, and camera aperture and skin regions (Supplementary Fig. 5g).

From the sclera region in the segmentation $\mathbf{M}_E$ we generate a refined sclera mask $\mathbf{M}_W$ from the image $I_{col}$ in CIELCh colour space. We exclude areas in the image with a lightness of $L^* < 50$ to remove dark features such as most eyelashes, we use a dilated mask of the iris segmentation $\mathbf{M}_I$ to remove the limbus, and we remove veins using a chroma and hue sensitive filter that removes pixels with chroma $C^* > 8$ and a reddish hue $h^* \in [345°, 60°]$. The masked sclera image $I_S$ is shown in Supplementary Fig. 5h.

## Sclera staining texture

To render a sclera staining texture, we extract a set $G$ of 9 sclera colours from the set of CIELAB colours of pixels of the masked sclera image $I_S$ using the k-means algorithm with 9 clusters and the k-means++[49] centre initialization (Supplementary Fig. 5h). We selected 9 clusters to allow sufficient colour variations while reducing the effect of

outliers. Based on the assumption that the staining causes the sclera to become darker, we determine the sclera base colour $\mathbf{g}_B$ as the average of the colours $\mathbf{g}_i \in G$

$$\mathbf{g}_B = \frac{1}{W}\sum_{i=1}^{9} w_i \mathbf{g}_i, \quad W = \sum_{i=1}^{9} w_i \tag{11}$$

weighted by the colours' lightness values $L_i^*$

$$w_i = \frac{\left(L_i^* - L_{\min}^*\right)}{\left(L_{\max}^* - L_{\min}^*\right)}, \quad L_{\min}^* = \min_{i\in\{1...9\}}\left(L_i^*\right), \quad L_{\max}^* = \max_{i\in\{1...9\}}\left(L_i^*\right) \tag{12}$$

and use this base colour $\mathbf{g}_B$ to create a uniform texture of width $u = 4096$ pixels and height $v = 2048$ pixels. On this texture we render stains for each colour $\mathbf{g}_i$ using weight maps generated using 3D Perlin noise[50] $N_P(x,y,z) : \mathbb{R}^3 \mapsto \mathbb{R}$ with cylindrical sampling to ensure matching seams at the vertical image borders in $v$. The weight maps are normalized, using histograms to compute the thresholds, such that the staining covers approximately 90% of the image and each stain colour occupies roughly the same area.

## Procedural vein network

Based on an approach from eye rendering[37] we procedurally grow a veining network in three layers (Supplementary Fig. 5j) using 15 vein recipes (5 recipes per layer). In a dimensionless 2D image with aspect ratio 2:1 we place the seed points for the first layer at 10 anatomically motivated positions. Each vein is simulated as list of nodes, its vein recipe determines the growth. The recipe influences in particular the values and variations of thickness, depth, length, branching and straightness of the vein. A vein stops growing after a fixed number of steps, if it becomes too thin, or once it reaches the limbus. Once grown veins branch into veins of the next layers at randomly selected nodes of the vein, a new vein starts at that node with a reduced thickness but same depth, however it has a different (randomly selected) vein recipe.

Two veining parameters thickness $th$ and branching ratio $br$ allow to influence the growth of the veining network, they directly scale the thickness and branching values of all vein recipes (Supplementary Fig. 5k). These veining parameters are selected by the ocularist for each patient using a set of reference prints (Supplementary Fig. 5p). For the 10 patients the thickness parameter $th$ was selected between 0.9 and 1.1, the branching ratio $br$ between 0.9 and 1.5.

## Veining texture rendering

After growing the network, the veins are rendered on top of the sclera texture, see Supplementary Fig. 5l, n. We use 15 manually sampled vein profiles, each contains a 1-dimensional colour and transparency cross-section with 21 values, which is sampled orthogonally to the vein's direction, and is assigned a depth and thickness. The vein profile is selected based on the minimal weighted squared distance to the vein's recipe depth and thickness, with a five times greater weight for the thickness.

We use quadratic B-Splines with four knots to rendering the vein segments between vein nodes, the distance to the curve is used to look up the colour in the vein profile (Supplementary Fig. 5m, o). The distance is locally modified with a sinusoidal offset perpendicular to the curve to simulate fine variations in the vein path. A depth map is used to correctly render overlapping veins, it can optionally be used to create a displacement map based on vein depth.

## Patient data acquisition

To capture the anophthalmic socket a conformer is selected in clinic with the pupil centre marked on the conformer window. The conformer is inserted into the patient's eye socket and the patient presents to the AS-OCT via a chin rest and forehead support frame.

The operator manually positions the AS-OCT head, with the crosshairs in the OCT software view, on the marked pupil centre or the window centre. A raster scan of the eye socket is captured, checked for correctness, and stored with a record of the conformer identifier.

Room lighting conditions are set for the imaging of the fellow eye based on a room lighting protocol. The AS-OCT device automatically focuses on the fellow eye. The patient blinks a few times then opens their eye wide before the AS-OCT device's light source is turned on to capture an unoccluded iris image, showing the full iris and as much of the sclera as possible. If the patient cannot open their eye wide enough, they can use their finger to pull down their lower lid and an assistant can hold the upper lid up with a long-tipped cotton bud. When the operator can see an unoccluded iris radial scan is captured he checks the iris mesh output, determines the patient's veining parameters, and records this in the patient dataset. The OCT machine requires 2.4 s to capture the raster scan of the eye socket or radial scan of the fellow eye.

## Prosthesis supply and adjustment

The prosthesis is fitted into the patient's socket and assessed for lid and iris positioning. Clear material from the front surface can be removed to close lids up; material from the rear fit surface can be removed to alter iris position and help reduce overall size if required. Tools used to do this are a Dremel type hand piece with tungsten carbide grinding bits, silicone smoothing/polishing wheels and calico polishing mops as well as a polishing lathe with calico mops and polishing compound.

Directly after each supply the cosmesis assessment was conducted in the same clinic room and by the same ocularist. The lighting in this room consisted of ceiling lights and some natural light from a window, the pictures were taken with an iPhone 13 Pro Max. After the assessment the patients left with the adjusted prosthesis as their new permanent ocular prosthesis.

## Data capture, processing, and analysis

The ocularist's grading of the prostheses were captured Microsoft Excel notebooks. The creation and analysis of the shape model as well as the computation and analysis of the colour characterization were performed using the colormath (3.0.0), matplotlib (3.7.0), numpy (1.24.4), and scikit-learn (1.1.1) Python packages. The data-driven design software uses OpenCV (4.3.0) for image processing and Eigen (3.3.1) for numerical minimizations.

## Patient population and recruitment

The reported patients were regular patients of MEH that required standard clinical care for an ocular prosthesis, which gave written informed consent to be supplied with a 3D printed prosthesis. IRB approval was obtained from the Medical Devices and New Technology Committee of Moorfields Eye Hospital and was registered by the Audit Department, number CA23/RE/960. The software was designed according to the requirements of IEC 62304 and self-registered with the MHRA as a class 1 medical device under UKCA.

The patients were asked during their supply visit whether they would also give written consent to be shown in a publication, of 17 patients asked 13 gave consent. However of those one was excluded because the supply was postponed, one because participation in a clinical trial, and one because the picture taken after supply was of poor quality.

The remaining patients, with the desired sample size of 10, were used for the reporting in this work.

## 3D printing and post-processing

All prostheses were printed by the printing service provider FIT AG in Lupburg, Germany on a Stratasys J750 (Stratasys Ltd., Israel) PolyJet 3D-printer, using Vero Pure White, Vero Black, Vero Clear, and Vero

Vivid colour materials at high mix mode with a resolution of 600, 300, and 940 dpi per axis. Slices for the mesh data were computed using the Cuttlefish 3D-printer driver (Fraunhofer IGD, Germany) in about 25 s per prosthesis; with an ICC colour profile specifically optimized for the iris and sclera colours, which was created with a proprietary colour profiler software (Fraunhofer IGD, Germany). Displacement maps are used to create a locally varying clear coating on the prosthesis surface. The slice data was sent to the 3D printer via the GrabCAD Voxel Print Utility. The print time per prosthesis depends strongly on the utilization of the print tray, while it takes about 90 min to print a single prosthesis it takes only about 10 h to print 100 prostheses at the same time. The printed prostheses were cleaned of support material and tumbled for 120 min after manufacture by the printing service provider FIT AG, scanning the results showed that this grinds off at most 0.3 mm of the shape, predominantly at the edges.

At the MEH the prostheses were then sonicated in the Branson Bransonic M-series (Thermo Fisher Scientific Inc., US) ultrasonic cleaning bath and polished by hand before given to the ocularist for the supply and adjustment to the patient.

### Material testing

To evaluate the biocompatibility of the used materials, material sample tests were conducted by Eurofins Biolab Srl. (Italy). The samples were produced and post-processed with the equipment and procedures described. The test results demonstrated that interpreted according to ISO 10993-5:2009 the samples were considered not cytotoxic, interpreted according to ISO 10993-11:2017 the samples did not cause systemic toxic symptoms in mice, interpreted according to ISO 10993-10:2010 the samples pass the test for intracutaneous reactivity in albino rabbits, and interpreted according to ISO 10993-10 the samples were considered not sensitizing in Guinea Pig models.

In addition, extracts in both polar (water) and non-polar (hexane) solvents of the same sample types were assessed via High-Performance Liquid Chromatography Ultraviolet Mass Spectrometry (HPLC-UV-MS), Gas Chromatography Mass Spectrometry (GC-MS), Headspace-Gas Chromatography/Mass Spectrometry (HS-GC/MS), and Inductively Coupled Plasma Mass Spectrometry (ICP-MS) to determine the presence of any extractable substances which might impact the biocompatibility of the device. All identified chemicals underwent a toxicological review and were determined to be within safe levels according to ISO 10993-17:2002.

### Reporting summary

Further information on research design is available in the Nature Portfolio Reporting Summary linked to this article.

## Data availability

The 3D scan data used for the statistical shape model is property of Ocupeye Ltd. and available for non-commercial use from the corresponding author on request. The patient image data is protected and not available due to data protection laws. An alternative example or synthetic input data set as well as an accompanying output 3D model is property of Ocupeye Ltd. and available from the corresponding author on request for non-commercial use.

## Code availability

The code for algorithms developed in this work is not available due to licensing agreements. The algorithms are provided as pseudocode in the Supplementary Information file and compiled binaries are available from the corresponding author upon request.

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

## Acknowledgements

We thank all patients. We thank Tomey Corp. for the modifications of the OCT device and software and Tomey GmbH for the provision of the OCT device. We thank A. Brunton for guidance on statistical shape models, Z. Paniwnyk for guidance on the software's certification, A. Whitton for providing details on the biocompatibility testing, A. Kraushaar for providing the colour patches, and N. Soliman for her help designing the report form. This project has received funding from the European Union's Horizon 2020 Eurostars programme under project E!113240 C2PAE (recipients J.R. and P.U.), and Moorfields Eye Charity and Drayson Foundation MEC GR001135 and GR001173 (recipient S.B. and M.S.S.). The research was supported by the National Institute for Health Research (NIHR) Biomedical Research Centre based at Moorfields Eye Hospital NHS Foundation Trust and UCL Institute of Ophthalmology. The views expressed are those of the author(s) and not necessarily those of the NHS, the NIHR or the Department of Health.

## Author contributions

S.B. and D.C. imaged the patients with the OCT device. S.B. scanned the prostheses with the 3D scanner. S.B. and J.R. measured the colour targets. P.U. advised on methods for the colour processing. D.C. prepared samples of iris and sclera colours. J.R. developed the methods for the shape scan processing, socket surface extraction, shape prediction and sclera texture generation. J.R. implemented the software and processed the data of the patients. S.B. and J.R. designed the conformers. S.B. cleaned and polished the prostheses. D.C. adjusted and supplied the prostheses. M.S.S. was responsible for the clinical implementation of the technology reported herein. J.R. drafted the manuscript and prepared the figures, all authors edited and approved the final version.

## Funding

## Competing interests

Fraunhofer - Gesellschaft zur Förderung der angewandten Forschung e.V. has filed a patent, EP22185166.0 (inventors J.R. and P.U.), that covers the data-driven design of the prosthesis. Ocupeye Ltd. has filed two patents, GB 2 586 629 and GB 2 589 698 (inventor S.B.), that covers the end-to-end process from scanning of patient to printing of eye and the general structure of the prosthesis. The authors declare no other competing interests.
