## [Peer Review File · Nature Communications]

Automatic data-driven design and 3D printing of custom ocular prosthesesREVIEWER COMMENTS

Reviewer #1 (Remarks to the Author):

- What are the noteworthy results? A novel process using data driven design and 3D printing to manufacture semi-custom prosthetic eyes.
- Will the work be of significance to the field and related fields? How does it compare to the established literature? If the work is not original, please provide relevant references. The work is very significant for the future of ocular prosthetics and related fields.
- Does the work support the conclusions and claims, or is additional evidence needed? Some of the benefits are overstated.
- Are there any flaws in the data analysis, interpretation and conclusions? Do these prohibit publication or require revision? Some revision required.
- Is the methodology sound? Does the work meet the expected standards in your field? Yes
- Is there enough detail provided in the methods for the work to be reproduced? Yes

COMMENTS

The authors should be thanked for investing considerable time and expertise in furthering the field of ocular prosthetics.

This review is limited to the aspects of the manuscript concerned with the practice of ocular prosthetics.

The manuscript is long and detailed and pitched at an audience that may not have as good an understanding of the subject as the authors. The manuscript would be much better if it was half the size and written for a particular audience – both ophthalmologists and bioengineers would be interested but the manuscript is confusing to both audiences in its present form. The order of photos in the figures is confusing and adds to the difficulty of reading the manuscript.

The use of the term 'custom ocular prostheses' in the title should be clarified. PMMA prosthetic eyes range from pre-manufactured 'stock' eyes (cheapest and most common) to 'modified stock' eyes (where stock eyes are reshaped to make them better fit individual sockets), to custom ocular prostheses. Custom ocular prostheses are full fitting with no dead spaces for lacrimal secretions to pool. Their anterior surfaces are moulded to match the curvature and lid contour of the companion eye and their iris/cornea components match the colour, texture, pupil size, iris diameter, iris position and direction of gaze of the companion eye. Scleral colours and veins also match. Custom ocular prostheses also fit irregular, atrophied or misshapen sockets.

The use of the term 'automatic' to describe the design and 3D printing process in the title overstates the simplicity of the process.

The authors correctly point out that current ocular prostheses vary considerably in quality between ophthalmologists and even between prostheses made by the same ophthalmologist. This is inevitable for a manual process and the authors should emphasise that reducing this inconsistency is a key benefit of the new technology.

No mention is made of scleral shell prostheses which are prosthetic eyes that fit over disfigured biological eyes with no useful vision. Scleral shell prostheses should be discussed because they might be more suited to 3D printing as scanning data from the ocular surface should be much more complete.

The discussion section overstates some of the benefits of the technology.

First: 'an intrusive and uncomfortable alginate impression is unnecessary.' Socket impressions taken with polyvinylsiloxane delivered via a mixing tip take a little over a minute to complete. Impressions taken in any material are seldom uncomfortable and no more intrusive than placing a conformer to scan the socket.

Third: '...allowing the ophthalmologist to see and supply about 5 times as many patients' This claim is subjective and depends on many unknown factors, including having a continuous supply of patients.

Fourth: '...allows uncomplicated supply of spares and replacements.' There is no need for this benefit other than for replacing a lost prosthesis. Ophthalmologists generally recommend replacing PMMA prosthetic eyes every 5 years or so. This is not because the prosthesis wears out but because they need to be updated as sockets change shape. A common consequence of not updating the prosthesis is a lax lower eyelid.

Reviewer #2 (Remarks to the Author):

This publication is noteworthy due to its advancement in automation. Prior literature has highlighted that we can transition from handmade products to 3D printed models. However, the work burden is not lessened by this change and the limiting factor (expertise), remains. This manuscript finds a way around this, creating a new path for improved products that can be made with much less expertise.

The methodology is sound and reproducible. As someone who has toiled over automated 3D printing designs, I can certainly attest to the importance of working this solution out and having the product be good enough for clinical use. Therefore, I recommend this manuscript be accepted for publication.

Reviewer #3 (Remarks to the Author):

An exhaustive and detailed study on the process for the digital and automatic manufacture of ocular prosthetics that fit the patients' eye sockets with contralateral symmetry. Clinically results appear promising and the approach requires reasonably less toil and time by the ocularist and produces reproducible output, however, cost effectivity must be clarified. But the authors have themselves mentioned that complex sockets still need the manufacturing craft of the ocularist for the shape and the the ocularist's skill is necessary for the final adjustment and fitting. Additionally it is not suitable for patients with certain eye conditions having nystagmus and strabismus.

So this signifies that Automatic data-driven design and 3D printing of custom ocular prostheses is beneficial for non complicated sockets which otherwise can be handled with the standard manual technique

It is reasonable that a new innovation should add inputs for the complicated situations which is otherwise difficult with the traditional ones. So it would be great if the authors can describe solution to the complicated sockets to aid significant inputs to the existing literature.

Reviewer #4 (Remarks to the Author):

It is necessary to produce customized prosthetic eyes to help social rehabilitation of anophthalmic patients, but it has limitations in that it is labor intensive and the results may vary depending on the capabilities of the ocularists. This study is meaningful as a comprehensive study that shows that this process can be simplified and standardized through data-driven design and 3D printing technology.

In this study, AS-OCT images taken while wearing a conformer are used to determine the shape of the prosthesis. However, the shape of the ocular prosthesis is mainly reflected by the patient's anophthalmic conjunctival sac except for its posterior surface (depth and shape of the superior and inferior sulcus, and the positional relationship between the fornix and orbital implant), while the AS-OCT only can scan the posterior surface of it.

1) For "shape model generation", 173 artificial eyes previously produced by a prosthetic eye company were used. Existing artificial eyes have the shape of an anophthalmic socket (which cannot be scanned with AS-OCT) through molding using alginate, so it is thought that the correlation between the posterior surface and anterior surface of the anophthalmic socket can be modeled. do. However, the process for estimating the remaining fornix and deriving the optimal anterior surface through posterior surface scanning using AS-OCT used in this study is not well described in the manuscript.

2) The shape of the prosthetic eye needs to be modified depending on the position (depth in orbit,

decentration) of the anophthalmic implant inserted during the patient's previous eye enucleation surgery. For example, in enophthalmic implants, the thickness of the ocular prosthesis must be thickened, and in patients with inferiorly decentered implants, the superior aspect of the ocular prosthesis must be reinforced. I wonder if these can be reflected in the automated process through the method described in this study.

3) For conformer modeling and generation, it was described as "the most common ocular prosthesis shapes, were scanned, aligned." It is suspected that the posterior surface of anophthalmic socket imaged with AS-OCT may be determined by the shape of the conformer. In particular, the fact that it says "The ocularist selects from a set of 12 conformers (Fig. 1h) the one that best fits in the patient's eye socket" for each patient raises the suspicion that there may be bias in the results depending on the choice of conformer. Can you show whether differences in results occur depending on the application of different conformers in the same patient?

We would like to thank the editor and the reviewers for their comments and suggestions. We have addressed all concerns in blue and improved our manuscript accordingly with changes highlighted in yellow (except for the reordering of the figures and updated Fig. 2 which is now Fig. 3).

REVIEWER COMMENTS

Reviewer #1 (Remarks to the Author):

- What are the noteworthy results? A novel process using data driven design and 3D printing to manufacture semi-custom prosthetic eyes.
- Will the work be of significance to the field and related fields? How does it compare to the established literature? If the work is not original, please provide relevant references. The work is very significant for the future of ocular prosthetics and related fields.
- Does the work support the conclusions and claims, or is additional evidence needed? Some of the benefits are overstated.
- Are there any flaws in the data analysis, interpretation and conclusions? Do these prohibit publication or require revision? Some revision required.
- Is the methodology sound? Does the work meet the expected standards in your field? Yes
- Is there enough detail provided in the methods for the work to be reproduced? Yes

COMMENTS

The authors should be thanked for investing considerable time and expertise in furthering the field of ocular prosthetics.

This review is limited to the aspects of the manuscript concerned with the practice of ocular prosthetics.

The manuscript is long and detailed and pitched at an audience that may not have as good an understanding of the subject as the authors. The manuscript would be much better if it was half the size and written for a particular audience – both ocularists and bioengineers would be interested but the manuscript is confusing to both audiences in its present form. The order of photos in the figures is confusing and adds to the difficulty of reading the manuscript.

We have also thought about splitting the publication into two publications with each having a deeper focus on the technical aspects and the (pre-) clinical results respectively. However, we believe that this topic provides a rare opportunity to present an interdisciplinary use of digital technology and additive manufacturing in a medical application that could spark more interest and collaboration between scientific disciplines that are otherwise unrelated. For that reason, we preferred to submit one manuscript.

We reordered the images in Fig.2, now Fig. 3, and hope that this makes the process clearer. We moved some details for the statistical shape model and conformer design from results text into method description to improve reading flow.

The use of the term 'custom ocular prostheses' in the title should be clarified. PMMA prosthetic eyes range from pre-manufactured 'stock' eyes (cheapest and most common) to 'modified stock' eyes (where stock eyes are reshaped to make them better fit individual sockets), to custom ocular prostheses. Custom ocular prostheses are full fitting with no dead spaces for lacrimal secretions to

pool. Their anterior surfaces are moulded to match the curvature and lid contour of the companion eye and their iris/cornea components match the colour, texture, pupil size, iris diameter, iris position and direction of gaze of the companion eye. Scleral colours and veins also match. Custom ocular prostheses also fit irregular, atrophied or misshapen sockets.

We clarified the term custom ocular prosthesis in the introduction. We have also clarified in the results that while the process intends to produce custom ocular prostheses the prostheses' shapes oftentimes still require adjustments, thus making them modified custom ocular prostheses.

The use of the term 'automatic' to describe the design and 3D printing process in the title overstates the simplicity of the process.

We acknowledge that this is not a fully automatic process but requires some manual labour (mostly in the final supply and adjustment session), however we think that the manual effort is reduced so much that terms like "semi-automatic" would not adequately describe the reduction. Furthermore, we believe that the process has the potential to improve with technical advances and thus steadily becomes more automated with each improvement. However, we have added a paragraph to explain the limitation of automation in the discussion section.

The authors correctly point out that current ocular prostheses vary considerably in quality between ocularists and even between prostheses made by the same ocularist. This is inevitable for a manual process and the authors should emphasise that reducing this inconsistency is a key benefit of the new technology.

We emphasised in the discussions that our process provides a consistent output that is very hard or impossible to achieve in the traditional process.

No mention is made of scleral shell prostheses which are prosthetic eyes that fit over disfigured biological eyes with no useful vision. Scleral shell prostheses should be discussed because they might be more suited to 3D printing as scanning data from the ocular surface should be much more complete.

While this process is currently tested on a small number of patients that need a scleral shell prosthesis, we do not yet have sufficient data to show any preliminary results. We added a sentence to the discussion that the suitability of our process for scleral shells is under evaluation in an ancillary study of the clinical trial.

The discussion section overstates some of the benefits of the technology.

First: 'an intrusive and uncomfortable alginate impression is unnecessary.' Socket impressions taken with polyvinylsiloxane delivered via a mixing tip take a little over a minute to complete. Impressions taken in any material are seldom uncomfortable and no more intrusive than placing a conformer to scan the socket.

Unfortunately, we can only share anecdotal evidence from the Moorfields Eye Hospital (MEH) where the observation was made that patients are more comfortable with the fitting of the wax shape than with the alginate impression. Recorded data of a clinical evaluation report for the 3D printed ocular conformer at the MEH demonstrates that no patient found the conformer uncomfortable, whereas, based on the ocularist's experience with the used traditional process, the majority of the patients perceive some form of discomfort in the socket during the alginate impression. However, since the experience can vary for the used process at a clinic, the ocularist, and each patient it is of course difficult to generalize. We rephrased to "possibly uncomfortable" as a compromise that reflects the experience at the MEH without asserting that this an issue for all patients.

Third: '...allowing the ocularist to see and supply about 5 times as many patients' This claim is subjective and depends on many unknown factors, including having a continuous supply of patients. We agree that this depends on many factors, and we have clarified that this is an observation from the Moorfields Eye Hospital and rephrased such that the ocularist could supply about 5 times more prostheses.

Fourth: '...allows uncomplicated supply of spares and replacements.' There is no need for this benefit other than for replacing a lost prosthesis. Ocularists generally recommend replacing PMMA prosthetic eyes every 5 years or so. This is not because the prosthesis wears out but because they need to be updated as sockets change shape. A common consequence of not updating the prosthesis is a lax lower eyelid.

This sentence was aimed on cases where the prosthesis has been lost and we have clarified this. Some patients have already asked for a set of spares in case they lose or damage a prosthesis (e.g. while swimming).

However, the frequency of the replacement of artificial eyes varies depending on the material and local health system. The glass eyes in Germany are replaced every two years because the surface becomes dull, for these types of prostheses it is also very difficult to reproduce the same (well fitting) shape in case the socket has not changed much because adjustments to the shape are more difficult and limited. We have heard from patients that still wear a very old prosthesis because they do not like the shape of their new prostheses. For these patients the ability to reproduce the shape of their old (comfortable) prosthesis can be very helpful.

Reviewer #2 (Remarks to the Author):

This publication is noteworthy due to its advancement in automation. Prior literature has highlighted that we can transition from handmade products to 3D printed models. However, the work burden is not lessened by this change and the limiting factor (expertise), remains. This manuscript finds a way around this, creating a new path for improved products that can be made with much less expertise.

The methodology is sound and reproducible. As someone who has toiled over automated 3D printing designs, I can certainly attest to the importance of working this solution out and having the product be good enough for clinical use. Therefore, I recommend this manuscript be accepted for publication.

We would like to thank the reviewer for their encouraging words.

Reviewer #3 (Remarks to the Author):

An exhaustive and detailed study on the process for the digital and automatic manufacture of ocular prosthetics that fit the patients' eye sockets with contralateral symmetry.

Clinically results appear promising and the approach requires reasonably less toil and time by the ocularist and produces reproducible output, however, cost effectivity must be clarified. \\

With respect to cost effectivity we acknowledge that, given that the machines that are necessary for this process are relatively expensive, it requires a certain scale to make it economically viable. While the process works out to be cost effective at a larger hospital such as the Moorfields Eye Hospital, allowing to offer the 3D printed prosthesis at the same price as a hand-made prosthesis, we are working towards a scalable solution that reduces the costs to make it also economically viable for smaller clinics or individual ocularists. We noted in the discussion that the process is cost effective once a certain volume is reached, but we would like to refrain from giving concrete numbers as these are business internals and depend on multiple variables.

But the authors have themselves mentioned that complex sockets still need the manufacturing craft of the ocularist for the shape and the the ocularist's skill is necessary for the final adjustment and fitting. Additionally it is not suitable for patients with certain eye conditions having nystagmus and strabismus.

So this signifies that Automatic data-driven design and 3D printing of custom ocular prostheses is beneficial for non complicated sockets which otherwise can be handled with the standard manual technique.

It is reasonable that a new innovation should add inputs for the complicated situations which is otherwise difficult with the traditional ones. So it would be great if the authors can describe solution to the complicated sockets to aid significant inputs to the existing literature.

As correctly identified there are some conditions that prevent patient from being supplied a prosthesis created with this process. We would like to point out that this report represents the beginning of this technology in prosthetic eyes and as mentioned we believe that continuous improvement of the software and hardware (OCT and printer) will allow more patients to benefit from the process in future. We envisage 2 possible scenarios in the medium term:

1. For patients with a complicated socket the ocularist can use a 3D scan of a manually made shape, in that case the ocularist still saves the effort to paint the prosthesis while the patient benefits from a generally better colour and appearance match.
2. For patients where the iris information cannot be captured with the OCT device, e.g. due to nystagmus, it could be possible to select from a digital database a matching iris using the (potentially blurry) colour calibrated image for reference.

We have mentioned the first case in the shape section of the results in the manuscript and noted in the discussion that software improvements can increase the number of eligible patients. However, we want to point out that even improvements for the non-complicated sockets can indirectly help patients with complicated sockets as the ocularist has more time left for care and treatment of those patients. Furthermore, we would like to note that complicated and difficult situations for the ocularist are not limited to the shape but also the appearance, the ocularist may struggle to manually replicate a certain iris colour and pattern that is easily reproducible with our process.

Reviewer #4 (Remarks to the Author):

It is necessary to produce customized prosthetic eyes to help social rehabilitation of anophthalmic patients, but it has limitations in that it is labor intensive and the results may vary depending on the capabilities of the ocularists. This study is meaningful as a comprehensive study that shows that this process can be simplified and standardized through data-driven design and 3D printing technology.

In this study, AS-OCT images taken while wearing a conformer are used to determine the shape of the prosthesis. However, the shape of the ocular prosthesis is mainly reflected by the patient's anophthalmic conjunctival sac except for its posterior surface (depth and shape of the superior and inferior sulcus, and the positional relationship between the fornix and orbital implant), while the AS-OCT only can scan the posterior surface of it.

1) For “shape model generation”, 173 artificial eyes previously produced by a prosthetic eye company were used. Existing artificial eyes have the shape of an anophthalmic socket (which cannot be scanned with AS-OCT) through molding using alginate, so it is thought that the correlation between the posterior surface and anterior surface of the anophthalmic socket can be modeled. However, the process for estimating the remaining fornix and deriving the optimal anterior surface through posterior surface scanning using AS-OCT used in this study is not well described in the manuscript. We understand that the generation of the shape’s anterior surface is not described well enough. We have expanded the description in results section of the manuscript to explain that the shape of the anterior surface and the fornix is the result of the shape fitting using the shape model. It is effectively a statistical prediction of how, based on the posterior surface of the anophthalmic socket, a likely fitting prosthesis shape, that fills the fornix and has a matching anterior surface, looks like.

2) The shape of the prosthetic eye needs to be modified depending on the position (depth in orbit, decentration) of the anophthalmic implant inserted during the patient's previous eye enucleation surgery. For example, in enophthalmic implants, the thickness of the ocular prosthesis must be thickened, and in patients with inferiorly decentered implants, the superior aspect of the ocular prosthesis must be reinforced. I wonder if these can be reflected in the automated process through the method described in this study.

It is possible to guide or steer the shape prediction via the choice of the conformer. For example the conformer, respectively it’s reference shape, could have some bulge at the superior area. Depending on the bias the predicted shape would be more or less influenced by the reference shape and contain its features. For enophthalmic sockets a thicker or deeper conformer could be used, which would then bias the shape prediction to produce a thicker prosthesis shape. We have clarified this in the shape section of the results in the manuscript.

3) For conformer modeling and generation, it was described as "the most common ocular prosthesis shapes, were scanned, aligned." It is suspected that the posterior surface of anophthalmic socket imaged with AS-OCT may be determined by the shape of the conformer. In particular, the fact that it says “The ocularist selects from a set of 12 conformers (Fig. 1h) the one that best fits in the patient’s eye socket” for each patient raises the suspicion that there may be bias in the results depending on the choice of conformer. Can you show whether differences in results occur depending on the application of different conformers in the same patient?

Supplementary Fig. 3h shows the differences in shape resulting from the bias towards different conformers for the same anophthalmic socket surface (of one patient). These differences are desired and allow the ocularist to influence or bias the shape prediction to the reference shape of the conformer. The selected conformer of course also influences the posterior anophthalmic socket, since

it may deform the tissue depending on the size. Unfortunately, we do not have such data for the patients shown in the manuscript, but for another patient (which we cannot show in the manuscript) we have one set of OCT scans with the A08 and A11 conformer inserted. Comparing these two conformers one can see that A11 is bigger than A08, which also visible affects the extracted socket surface from the OCT scan. In the scan for A11 the surface is pushed deeper into the socket, leading to a deformed socket surface. Because of the too big conformer the resulting shape was not a good fit for that patient. The next scan used the A08 conformer, which is smaller; this time the socket surface was not deformed, and the resulting prosthesis was successfully fitted and supplied to the patient. We have clarified in the results section of the manuscript that the ocularist must select a conformer that matches the posterior anophthalmic socket surface and socket volume well, since too big or too small conformers will lead to a deformed socket surface and provide an erroneous socket surface.

REVIEWERS' COMMENTS

Reviewer #1 (Remarks to the Author):

Thank you for your thoughtful responses to the reviewers' comments.

This paper goes a long way towards establishing 3D printing as a viable process to overcome the wide variability inherent in manual prosthetic eye manufacture and fitting.

I am happy for your manuscript to be published and look forward to further development of the technology in the future.

Reviewer #3 (Remarks to the Author):

The description of Automatic data-driven design and 3D printing of custom ocular prostheses is well written and the authors have done the necessary correction with exhaustive explanation.

Reviewer #4 (Remarks to the Author):

The issues raised were appropriately addressed.

We would like to thank the editor and the reviewers again for their comments and suggestions that helped us to improve our manuscript.

REVIEWER COMMENTS

Reviewer #1 (Remarks to the Author):

Thank you for your thoughtful responses to the reviewers' comments.

This paper goes a long way towards establishing 3D printing as a viable process to overcome the wide variability inherent in manual prosthetic eye manufacture and fitting.

I am happy for your manuscript to be published and look forward to further development of the technology in the future.

Reviewer #3 (Remarks to the Author):

The description of Automatic data-driven design and 3D printing of custom ocular prostheses is well written and the authors have done the necessary correction with exhaustive explanation.

Reviewer #4 (Remarks to the Author):

The issues raised were appropriately addressed.